# Discovering Quality-Diversity Algorithms via Meta-Black-Box Optimization

## Abstract

Quality-Diversity has emerged as a powerful family of evolutionary algorithms that generate diverse populations of high-performing solutions by implementing local competition principles inspired by biological evolution. While these algorithms successfully foster diversity and innovation, their specific mechanisms rely on heuristics, such as grid-based competition in MAP-Elites or nearest-neighbor competition in unstructured archives. In this work, we propose a fundamentally different approach: using meta-learning to automatically discover novel Quality-Diversity algorithms. By parameterizing the competition rules using attention-based neural architectures, we evolve new algorithms that capture complex relationships between individuals in the descriptor space. Our discovered algorithms demonstrate competitive or superior performance compared to established Quality-Diversity baselines while exhibiting strong generalization to higher dimensions, larger populations, and out-of-distribution domains like robot control. Notably, even when optimized solely for fitness, these algorithms naturally maintain diverse populations, suggesting meta-learning rediscovers that diversity is fundamental to effective optimization.

## 1 Introduction

Over billions of years, biological evolution has created the extraordinary complexity of life on Earth, giving rise to the flight of birds, the computational power of the human brain, and the collaborative intelligence of ant colonies. This remarkable process inspired the development of Genetic Algorithms (Holland, 1992) (GA), which replicate the essential principles of biological evolution in silico to solve challenging computational problems. In their classic implementation, GAs maintain a population of candidate solutions that evolve over generations through reproduction, evaluation and selection. However, this process typically implements selection through global competition, where each individual competes against the entire population for survival. Such a population-wide selection pressure often leads to premature convergence toward a single suboptimal solution, losing the diversity that makes natural evolution so powerful (Mouret & Clune, 2015).

In contrast, natural evolution is driven by local competition, where individuals compete primarily with neighbors sharing the same environment (Darwin, 1859). This localized competitive pressure naturally leads to the emergence of diverse specialized organisms. When success is defined by outperforming immediate neighbors rather than the entire population, species can adapt to their specific niches without being eliminated by globally superior solutions. Each local environment presents unique challenges and opportunities, fostering specialized traits optimal for that context, rather than forcing convergence to a single dominant species.

The power of local competition in driving both diversity and innovation has been harnessed in Quality-Diversity algorithms (Pugh et al., 2016) (QD), a family of methods that evolve populations of diverse, high-performing solutions. Notably, maintaining diversity has been shown to improve not just population-level adaptation but also peak performance: by promoting exploration and serendipitous discoveries, the best solutions emerge from diverse populations rather than pure fitness optimization (Stanley & Lehman, 2015; Faldor et al., 2023; 2024). This counter-intuitive benefit arises because preserving diverse solutions, even when initially suboptimal, creates crucial stepping stones for evolution. These intermediate solutions can later evolve into high-performing variants that would have been inaccessible through greedy optimization.

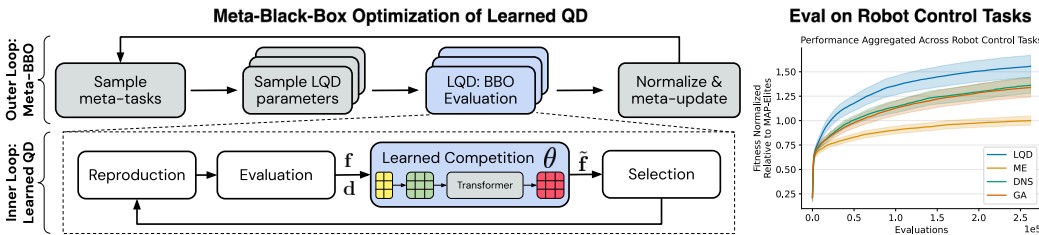

Figure 1: Meta-black-box optimization of Learned Quality-Diversity (LQD) algorithms. At each meta-generation, a meta-evolution strategy (ES) samples candidate LQD parameters and evaluates them on sampled black-box optimization tasks. The LQD algorithm is run, performance is standardized across tasks, and the meta-ES is updated accordingly. After meta-optimization the discovered LQD outperforms various baselines across 6 aggregated *out-of-distribution* robot control tasks.

To implement local competition, QD algorithms typically embed individuals in a vector space — called the descriptor space — where distances between vectors determine which solutions compete with each other and what constitutes a novel solution. MAP-Elites (Mouret & Clune, 2015) (ME), a prominent QD algorithm, implements local competition by discretizing this descriptor space into a grid of cells, where competition occurs only between solutions mapped to the same cell. Alternative approaches maintain unstructured archives of solutions where competition occurs between similar individuals without imposing a rigid spatial organization (Lehman & Stanley, 2011a; Faldor & Cully, 2024; Cully, 2019; Bahlous-Boldi et al., 2024). While these Quality-Diversity algorithms successfully implement local competition, their particular mechanisms often rely on heuristics. For instance, ME's grid-based competition rely on a rather crude discretization of the descriptor space (Bahlous-Boldi et al., 2024; Kent et al., 2022; Vassiliades et al., 2017), while unstructured archives often rely on additional hyperparameters that prove particularly challenging to tune (Grillotti & Cully, 2022).

In this work, we explore a fundamentally different approach: given a sufficiently flexible parameterization of the competition rules, we discover new Quality-Diversity algorithms using meta-black-box optimization (Lange et al., 2023a;b; 2024). Our key contributions include: (1) Learned Quality-Diversity (LQD), a flexible framework that parameterizes competition rules using attention-based neural architectures, enabling the discovery of sophisticated Quality-Diversity algorithms; (2) empirical evidence that our learned algorithms match or exceed the performance of established baselines like MAP-Elites (Mouret & Clune, 2015) and Dominated Novelty Search (Bahlous-Boldi et al., 2024); and (3) evidence of strong generalization capabilities, with LQD scaling effectively to higher dimensions, larger populations, and novel domains like robot control tasks. Our results suggest that meta-learning can automatically discover sophisticated local competition strategies that advance the state-of-the-art in Quality-Diversity optimization.

## 2 BACKGROUND AND RELATED WORK

For a full discussion of related works see Appendix A. While the Meta-Black-Box Optimization (Meta-BBO) landscape includes recent state-of-the-art advancements (e.g., (Ma et al., 2025; Guo et al., 2025; Li et al., 2024), see Appendix A for a comprehensive discussion), we adopt the established framework of Lange et al. (2023b) for simplicity. This allows us to isolate our primary contribution: the parameterization and discovery of novel Quality-Diversity competition rules.

### 2.1 BLACK-BOX OPTIMIZATION

Black-box optimization (BBO) addresses the fundamental challenge of finding optimal solutions without access to gradients or internal function structure. In this work, we focus on real-parameter black-box optimization. Given an objective function $f : \mathbb{R}^n \to \mathbb{R}$ with unknown functional form, we seek to solve: $\max_{\mathbf{x} \in \mathcal{X}} f(\mathbf{x})$ where the search space $\mathcal{X} \subset \mathbb{R}^n$ is a $n$-dimensional box.

### 2.2 GENETIC ALGORITHMS

Genetic Algorithms (Holland, 1992; Rechenberg & Eigen, 1973) (GAs) are population-based black-box optimization methods that evolve a population of candidate solutions across generations. These

algorithms have become fundamental tools in optimization due to their ability to handle non-convex, multi-modal landscapes where traditional gradient-based methods often fail. Algorithm 1 outlines a standard GA implementation, though numerous variations exist.

At each generation, a GA updates a population of $N$ solutions $\mathbf{X} = (\mathbf{x}_i)_{i=1}^N \in \mathbb{R}^{N \times n}$ with corresponding fitness values $\mathbf{f} = (f_i)_{i=1}^N \in \mathbb{R}^N$. First, $B$ new offspring solutions $\mathbf{X}' = $ REPRODUCTION$(\mathbf{X}, \mathbf{f})$ are generated through mechanisms like crossover and mutation. These offspring are added to the population, creating a set CONCAT$(\mathbf{X}, \mathbf{X}')$ of size $N+B$. Then,

---

**Algorithm 1** Genetic Algorithm

**Require:** population size $N$, reproduction batch size $B$
  Initialize population $\mathbf{X}$ with fitness $\mathbf{f}$
  **for** each generation **do**
      $\mathbf{X}' \leftarrow$ REPRODUCTION$(\mathbf{X}, \mathbf{f})$    ▷ Generate $B$ offspring
      $\mathbf{X} \leftarrow$ CONCAT$(\mathbf{X}, \mathbf{X}')$    ▷ Add offspring to population
      $\mathbf{f} \leftarrow$ EVALUATION$(\mathbf{X})$    ▷ Evaluate fitness
      $\tilde{\mathbf{f}} \leftarrow$ COMPETITION$(\mathbf{f})$    ▷ Global competition (identity)
      $\mathbf{X} \leftarrow$ SELECTION$(\mathbf{X}, \tilde{\mathbf{f}})$    ▷ Keep top-$N$ individuals

---

COMPETITION computes the *competition fitness* $\tilde{\mathbf{f}}$ that will determine survival. In classic GAs, this function is simply the identity function $\tilde{\mathbf{f}} = \mathbf{f}$, meaning individuals engage in global competition based on their raw fitness values — this baseline will be modified in QD algorithms to implement local competition. Finally, the selection operator ranks all solutions according to their competition fitness $\tilde{\mathbf{f}}$ and retains only the top-$N$ solutions through truncation, forming the next generation.

## 2.3 QUALITY-DIVERSITY

Quality-Diversity (Pugh et al., 2016) is a family of genetic algorithms that implement local competition to generate diverse, high-performing solutions. In a QD algorithm, each solution is characterized by both its fitness value $f$ and a descriptor vector $\mathbf{d} \in \mathcal{D} \subset \mathbb{R}^D$ that captures meaningful features of the individual. The descriptors define precisely how solutions differ from each other, providing a mathematical foundation for local competition where solutions compete primarily with similar neighbors rather than the entire population.

At each generation, a QD algorithm updates a population of $N$ solutions $\mathbf{X} = (\mathbf{x}_i)_{i=1}^N \in \mathbb{R}^{N \times n}$ with corresponding fitness values $\mathbf{f} = (f_i)_{i=1}^N \in \mathbb{R}^N$ and descriptors $\mathbf{d} = (\mathbf{d}_i)_{i=1}^N \in \mathbb{R}^{N \times D}$. The key innovation lies in replacing global competition with local competition. While GAs used the identity function for COMPETITION, QD algorithms compute competition

---

**Algorithm 2** Quality-Diversity

**Require:** population size $N$, reproduction batch size $B$
  Initialize population $\mathbf{X}$ with fitness $\mathbf{f}$ and descriptors $\mathbf{d}$
  **for** each generation **do**
      $\mathbf{X}' \leftarrow$ REPRODUCTION$(\mathbf{X}, \mathbf{f})$    ▷ Generate $B$ offspring
      $\mathbf{X} \leftarrow$ CONCAT$(\mathbf{X}, \mathbf{X}')$    ▷ Add offspring to population
      $\mathbf{f}, \mathbf{d} \leftarrow$ EVALUATION$(\mathbf{X})$    ▷ Evaluate fitness and descriptor
      $\tilde{\mathbf{f}} \leftarrow$ COMPETITION$(\mathbf{f}, \mathbf{d})$    ▷ Local competition
      $\mathbf{X} \leftarrow$ SELECTION$(\mathbf{X}, \tilde{\mathbf{f}})$    ▷ Keep top-$N$ individuals

---

fitness values $\tilde{\mathbf{f}}$ based on both $\mathbf{f}$ and $\mathbf{d}$. Individuals primarily compete with others having similar descriptors, creating local competitive pressures that promote both quality and diversity. The selection of individuals for the next generation remains the same — ranking by $\tilde{\mathbf{f}}$ followed by truncation — but now operates on locally-modified fitness values. This local competition allows the population to maintain diverse solutions adapted to different niches while driving improvement within each niche.

### 2.3.1 MAP-ELITES

ME is a prominent QD algorithm that implements local competition by discretizing the descriptor space into a grid of cells (Mouret & Clune, 2015; Vassiliades et al., 2018). Each cell in the grid is represented by a centroid $\mathbf{c}_i$ and maintains at most one solution. Competition occurs only between solutions mapped to the same cell. Given a population with fitness $\mathbf{f}$ and descriptors $\mathbf{d}$, the COMPETITION function operates as follows:

1. Each individual is assigned to its nearest centroid based on its descriptor, partitioning the population into grid cells.

2. Within each cell $i$, only the highest-fitness individual $\mathbf{x}_k$ maintains its original fitness value $\tilde{f}_k = f_k$, while all other individuals $\mathbf{x}_j$ are eliminated by setting $\tilde{f}_j = -\infty$, for $j \neq k$.

This mechanism ensures that only the best-performing solution survives in each cell of the descriptor space, effectively implementing local competition.

### 2.3.2 NOVELTY SEARCH

While not strictly a QD algorithm, Novelty Search (NS) pioneered key ideas that influenced QD approaches (Lehman & Stanley, 2011a). This algorithm rewards solutions for being different from an archive of past solutions. Here, we present a simplified variant where novelty is computed with regards to the current population. Given a population with fitness $\mathbf{f}$ and descriptors $\mathbf{d}$, the COMPETITION function operates as follows:

1. For each solution $\mathbf{x}_i \in \mathbf{X}$, compute the descriptor space distances to *all* other solutions $\mathcal{D}_i = \{\|\mathbf{d}_i - \mathbf{d}_j\| \,|\, j = 1, \ldots, N, j \neq i\}$.
2. Compute the *novelty score*, $n(\mathbf{x}_i)$, which is the average distance to the $k$-nearest neighbors in $\mathcal{D}_i$. Set the competition fitness $\tilde{f}_i = n(\mathbf{x}_i)$.

This mechanism encourages solutions to explore novel regions of the descriptor space that are distant from other solutions.

### 2.3.3 DOMINATED NOVELTY SEARCH

Dominated Novelty Search (Bahlous-Boldi et al., 2024) is a QD algorithm that implements local competition using COMPETITION function that rewards solutions for being different from their *fitter* neighbors. Given a population with fitness $\mathbf{f}$ and descriptors $\mathbf{d}$, the COMPETITION function operates as follows:

1. For each solution $\mathbf{x}_i \in \mathbf{X}$, compute the descriptor space distances to *fitter* solutions $\mathcal{D}_i = \{\|\mathbf{d}_i - \mathbf{d}_j\| \,|\, j = 1, \ldots, N, j \neq i, f_j > f_i\}$.
2. Compute the *dominated novelty score*, $n^+(\mathbf{x}_i)$, which is the average distance to the $k$-nearest-fitter neighbors. Set the competition fitness $\tilde{f}_i = n^+(\mathbf{x}_i)$.

This mechanism encourages solutions to explore regions of the descriptor space that are distant from better-performing solutions, naturally balancing quality and diversity.

## 3 METHOD

In this section, we present our approach to discovering novel QD algorithms through Meta-Black-Box Optimization. First, we introduce Learned Quality-Diversity (LQD), a flexible framework that can represent any QD algorithms by parameterizing its competition function (Section 3.1). Then, we detail the meta-learning procedure used to discover effective LQD algorithms (Section 3.2).

### 3.1 LEARNED QUALITY-DIVERSITY ALGORITHM

As shown in Algorithms 1 and 2, Quality-Diversity algorithms can be viewed as Genetic Algorithms with local competition replacing global competition. The key difference lies in the COMPETITION function: while GAs use identity mapping (global competition), QD algorithms implement various forms of local competition. For instance, ME (Section 2.3.1) uses grid-based competition, while Dominated Novelty Search (Section 2.3.3) competes with nearest fitter neighbors. LQD generalizes this framework by parameterizing the COMPETITION function as a transformer with parameters $\theta$. Algorithm 3 outlines the complete LQD algorithm.

---

**Algorithm 3** Learned Quality-Diversity

**Require:** population size $N$, reproduction batch size $B$, params $\theta$
  Initialize population $\mathbf{X}$ with fitness $\mathbf{f}$ and descriptors $\mathbf{d}$
  **for** each generation **do**
    $\mathbf{X}' \leftarrow$ REPRODUCTION$(\mathbf{X}, \mathbf{f})$       ▷ Generate $B$ offspring
    $\mathbf{X} \leftarrow$ CONCAT$(\mathbf{X}, \mathbf{X}')$       ▷ Add offspring to population
    $\mathbf{f}, \mathbf{d} \leftarrow$ EVALUATION$(\mathbf{X})$   ▷ Evaluate fitness and descriptor
    $\tilde{\mathbf{f}} \leftarrow$ COMPETITION$_\theta(\mathbf{f}, \mathbf{d})$   ▷ Learned local competition
    $\mathbf{X} \leftarrow$ SELECTION$(\mathbf{X}, \tilde{\mathbf{f}})$       ▷ Keep top-$N$ individuals

---

We choose the transformer architecture (Vaswani et al., 2017) for two key properties that are crucial in our context. First, transformers are permutation equivariant - the output for each individual depends

on its relationship with the whole population, regardless of how individuals are ordered. This property is essential for evolutionary algorithms where the ordering of individuals in the population should not affect the competition. Second, transformers represent an architecture with minimal inductive bias, making them highly expressive and capable of representing virtually any competition rule.

The learned competition function processes the population's fitness and descriptor through several transformations (Figure 2):

1. **Featurize**: Fitness values $\mathbf{f}$ and descriptors $\mathbf{d}$ are concatenated and standardized across the population to form a matrix $\mathbf{z} \in \mathbb{R}^{N \times (D+1)}$.

2. **Embed**: The standardized features $\mathbf{z}$ are projected into a higher-dimensional space to form embeddings suitable for transformer processing.

3. **Transformer**: These embeddings are processed by a transformer network (Vaswani et al., 2017) that maintains permutation equivariance, ensuring the competition rules remain consistent regardless of population ordering.

4. **Output projection**: The transformer's output ($\mathbb{R}^{N \times D_K}$) is projected to scalar values ($\tilde{\mathbf{f}} \in \mathbb{R}^N$) that determine survival in the selection step.

Notably, we do not embed seemingly fundamental evolutionary principles into the architecture — even the common practice of maintaining the highest-fitness solutions in the population must be discovered if it proves beneficial. Through meta-black-box optimization, this flexible architecture allows us to explore the vast space of possible Quality-Diversity competition rules with minimal constraints, effectively searching through all archive mechanisms.

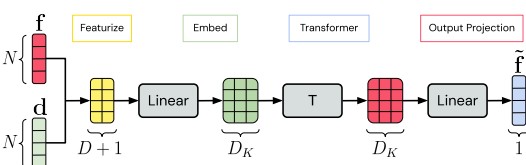

Figure 2: Competition function architecture.

### 3.2 META-BLACK-BOX OPTIMIZATION

To discover effective LQD algorithms, we adapt the meta-learning procedure introduced by Lange et al. (2023b) for evolving evolutionary algorithms. Our meta-optimization approach consists of three key components: a diverse set of optimization problems that form the meta-task (Section 3.2.1), three meta-objectives (Section 3.2.2), and an efficient meta-training procedure (Section 3.2.3).

#### 3.2.1 META-TASK

We train LQD on a diverse distribution of 22 BBOB (Black-Box Optimization Benchmark) functions from Finck et al. (2010a). These functions encompass a wide range of optimization challenges, including varying degrees of separability, conditioning, and multi-modality, ensuring our learned algorithm develops robust strategies. The complete set of training functions is detailed in Table 6. To further enhance the robustness of our LQD algorithm, we incorporate different noise models from Finck et al. (2010b) during training, including uniform, Gaussian, and Cauchy noise distributions. This noise injection helps ensure the learned competition rules remain effective even under noisy fitness evaluations.

During meta-training, we employ extensive data augmentation on the BBO tasks to improve generalization. Specifically, BBOB functions are sampled with varying dimensionality (from 2 to 12 dimensions), different noise model parameters, and random rotations of the search space. This augmentation strategy creates a rich training distribution that helps the learned algorithm develop robust competition rules that generalize across problems.

Finally, to transform these BBO tasks into QD tasks, we associate each task with a descriptor space through random projection. When sampling a task during meta-training, we generate a random matrix $\mathbf{N} \in \mathbb{R}^{D \times n}$ with independent standard normal entries and compute descriptors for each solution $i$ as $\mathbf{d}_i = \mathbf{N}\mathbf{x}_i$. In standard QD settings, the evaluation process typically involves a mapping from genotype to phenotype, and subsequently from phenotype to descriptor. While this composite mapping is often highly complex and non-linear — such as computing the trajectory of a robot from

its controller parameters — it nevertheless remains a function mapping the genotype to the descriptor. In this work, we simulate this mapping using a simple linear random projection from genotype to descriptor. Our empirical results demonstrate that meta-training on this simplified linear mapping is sufficient to learn competition rules that generalize to complex tasks, such as robot control tasks involving intricate trajectory-based descriptors.

### 3.2.2 META-OBJECTIVE

We define three distinct meta-objectives to evolve specialized LQD variants. Each meta-objective $\mathcal{L}(\theta_i)$ evaluates a candidate algorithm with parameters $\theta_i$ across a meta-batch of $K$ tasks $(\xi_k)_{k=1}^K$. The meta-objective is computed following Lange et al. (2023b). First, for each parameters $\theta_i$ and each task $\xi_k$, we compute a raw score, $s(\theta_i, \xi_k)$. The formula for this score depends on the specific meta-objective, as detailed in Table 1. Second, to enable fair comparison across different tasks, we standardize the raw scores. For a given task $\xi_k$, we compute the z-score of $s(\theta_i, \xi_k)$ across all candidate algorithms $(\theta_i)_{i=1}^M$ in the meta-population. This standardized score, denoted $\hat{s}(\theta_i, \xi_k)$, measures how well algorithm $\theta_i$ performed on task $\xi_k$ relative to others. Finally, the final fitness for algorithm $\theta_i$ is the median of its standardized scores across all tasks: $\mathcal{L}(\theta_i) = \text{median}_k \hat{s}(\theta_i, \xi_k)$.

Table 1: Meta-objectives used to evolve specialized LQD variants. The raw score is used to compute the meta-objective during training, while the eval metric is used to evaluate the algorithms in Section 4.

| Name | Variant | Raw score | Meta-objective | Eval metric |
|---|---|---|---|---|
| Quality | LQD (Q) | $s_Q(\theta_i, \xi_k) = \max_{\mathbf{x} \in \mathbf{X}} f(\mathbf{x})$ | $\mathcal{L}(\theta_i) = \underset{k \in \{1,..,K\}}{\text{median}} \hat{s}_Q(\theta_i, \xi_k)$ | $\max_{\mathbf{x} \in \mathbf{X}} f(\mathbf{x})$ |
| Diversity | LQD (D) | $s_D(\theta_i, \xi_k) = \underset{\mathbf{x} \in \mathbf{X}}{\text{mean}}\, n(\mathbf{x})$ | $\mathcal{L}(\theta_i) = \underset{k \in \{1,..,K\}}{\text{median}} \hat{s}_D(\theta_i, \xi_k)$ | $\underset{\mathbf{x} \in \mathbf{X}}{\text{mean}}\, n(\mathbf{x})$ |
| QD | LQD (QD) | $\begin{aligned} s_Q(\theta_i, \xi_k) &= \underset{\mathbf{x} \in \mathbf{X}}{\text{mean}}\, f(\mathbf{x}) \\ s_D(\theta_i, \xi_k) &= \underset{\mathbf{x} \in \mathbf{X}}{\text{mean}}\, n(\mathbf{x}) \end{aligned}$ | $\mathcal{L}(\theta_i) = \underset{k \in \{1,..,K\}}{\text{median}}\, \alpha \hat{s}_Q + (1-\alpha)\hat{s}_D$ | $\underset{\mathbf{x} \in \mathbf{X}}{\text{mean}}\, n^+(\mathbf{x})$ |

The Quality objective is designed to discover pure optimizers. The raw score is the single highest fitness value achieved in the final population, rewarding peak performance. The Diversity objective encourages broad exploration of the descriptor space. The raw score is the average novelty of the final population, rewarding the discovery of diverse solutions. Finally, the Quality-Diversity objective balances both quality and diversity. The final objective is a weighted combination of the normalized mean fitness and normalized mean novelty scores. The last column, evaluation metric, indicates the metric used to benchmark each variant in Section 4. Notably, for LQD (QD), we evaluate using the mean dominated novelty $n^+$, rather than the weighted combination used for meta-training.

### 3.2.3 META-TRAINING

We optimize the LQD parameters on a diverse family of BBO tasks described in Section 3.2.1, following the Meta-BBO procedure established in previous work (Lange et al., 2023b;a; 2024). Each set of parameters $\theta_i$ defines a unique QD algorithm through the competition function COMPETITION$_{\theta_i}$ by specifying how solutions compete with each other. The meta-training process, illustrated in Figure 1, maintains a meta-population of $M = 512$ LQD parameters $\theta_i$ for $i = 1, ..., M$. At each meta-generation, we uniformly sample a set of $K = 512$ BBO tasks $\xi_k$

---

**Algorithm 4** Meta-Black-Box Optimization of Learned QD

---

**Require:** Meta-ES, meta-population size $M$, meta-task with meta-batch size $K$, population size $N$, reproduction batch size $B$
    Initialize meta-ES distribution $(\mu, \Sigma)$
    **while** not done **do**
        Sample $K$ BBO tasks $\xi_k, \forall k = 1, \ldots, K$
        Sample $M$ LQD parameters $\theta_i \sim \mathcal{N}(\mu, \Sigma), \forall i = 1, \ldots, M$
        Evaluate the M LQD parameters on the same $K$ tasks:
        **for** $i = 1, \ldots, M$ **do**
            **for** $k = 1, \ldots, K$ **do**
                Inner loop to get $\left[(\mathbf{x}_{j,t})_{j=1}^N\right]_{t=1}^T$ (Algorithm 3)
        Collect populations at final step $\left[\left[(\mathbf{x}_{j,T})_{j=1}^N \Big| \xi_k\right]_{k=1}^K \Big| \theta_i\right]_{i=1}^M$
        Compute meta-objective $(\mathcal{L}(\theta_i))_{i=1}^M$ (Section 3.2.2)
        Update distribution $(\mu, \Sigma) \leftarrow \text{Meta-ES}\left((\mathcal{L}(\theta_i))_{i=1}^M \big| \mu, \Sigma\right)$

---

for $k = 1, ..., K$ and evaluate each LQD variant across all tasks. These evaluations are aggregated into meta-objectives that guide the meta-evolutionary optimization.

The LQD parameters are trained for 16,384 meta-generations using SNES (Wierstra et al.), with each inner loop running for 256 generations with a population size of 128. After meta-training, this results in three specialized LQD variants depending on the meta-objective, see Section 3.2.2. This process is effectively a search through the space of possible Quality-Diversity algorithms to discover competition rules that are optimized for different objectives.

The complete meta-optimization process is detailed in Algorithm 4, with comprehensive hyperparameter settings provided in Table 5. Notably, the transformer architecture uses 6 layers with 16 features and 4 attention heads, striking a balance between expressivity and computational efficiency.

## 4 EXPERIMENTS

We conduct an extensive empirical evaluation of our meta-optimization procedure and the discovered Learned QD algorithms. Our experiments are designed to answer three key research questions:

- Can we meta-learn QD algorithms that demonstrate competitive performance against established baselines on BBOB functions (Section 4.2)?
- How do these learned algorithms generalize to fundamentally different domains like robot control tasks, with higher dimensionality and domain-specific descriptors (Section 4.3)?
- What competition strategies emerge when optimizing different meta-objectives, and how do these learned mechanisms relate to existing approaches (Section 5)?

For statistical rigor, we conducted 32 independent replications of each experiment using distinct seeds. Statistical significance is assessed using the Wilcoxon–Mann–Whitney $U$ test with Holm-Bonferroni correction. Our implementation leverages hardware acceleration through JAX (Bradbury et al., 2018), with optimization algorithms from evosax (Lange, 2022) and QDax (Chalumeau et al., 2024). All experiments were conducted on 8xH100 GPUs.

### 4.1 BASELINES

We compare LQD against five established baseline algorithms. MAP-Elites (Section 2.3.1) is a QD algorithm that partitions solutions into grid cells for local competition. Dominated Novelty Search (Section 2.3.3) is another QD algorithm, that uses nearest-neighbor local competition. We also include a traditional Genetic Algorithm (Section 2.2) that employs global competition. A further baseline is Novelty Search (Section 2.3.2), implemented as a variant where novelty scores are computed using only the current population. Finally, we also use a Random baseline for normalizing performance, which is a GA using a random ranking during selection.

### 4.2 BLACK-BOX OPTIMIZATION BENCHMARK TASKS

We evaluate the performance of our three LQD variants against established baselines across both the meta-training tasks (Table 6) and a set of challenging out-of-distribution optimization problems (Table 7). To ensure a strictly fair comparison, all algorithms were run with identical hyperparameters regarding computational cost: a population size of 128 and a fixed budget of $1,000,000$ evaluations. The three LQD models are trained once, and used without any task-specific fine-tuning across all experiments. For out-of-distribution BBO evaluation, we selected six complex functions: Gallagher's Gaussian 101-me Peaks Function and Gallagher's Gaussian 21-hi Peaks Function from Finck et al. (2010a), as well as the Ackley, Dixon-Price, Griewank and Levy functions from Jamil & Yang (2013).

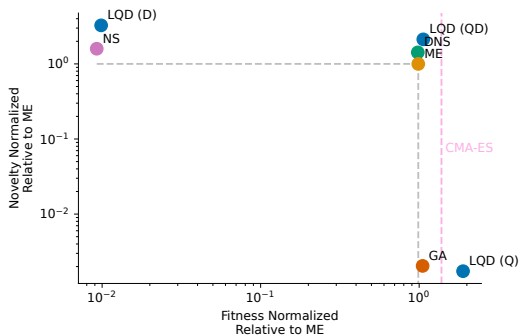

Figure 3: Quality-Diversity trade-off for each algorithm. Each point represents the median fitness and novelty scores achieved across 32 runs on BBO training tasks after one million evaluations.

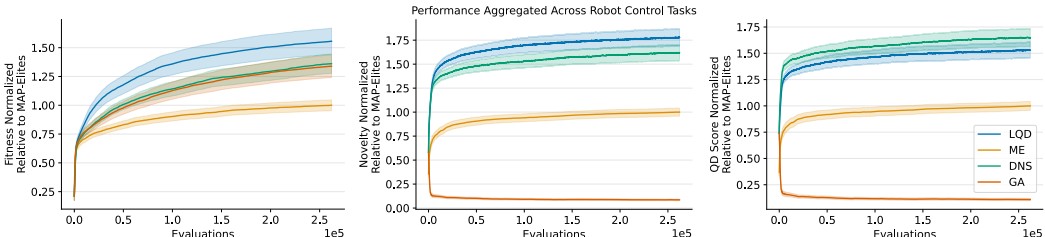

Figure 4: Performance across robot control tasks for LQD (Q) (left), LQD (D) (middle) and LQD (QD) (right) compared to baselines. Lines show mean performance across 32 independent runs, with shaded regions indicating 95% confidence intervals. LQD demonstrates strong generalization, matching or exceeding baseline performance despite never being trained on robotic tasks.

We conducted extensive experiments to evaluate the performance of our three LQD variants against established baselines across both the meta-training tasks (Table 6) and a set of challenging out-of-distribution optimization problems (Table 7). The three LQD models are trained once, and used without any task-specific fine-tuning across all experiments. For out-of-distribution BBO evaluation, we selected six complex functions: Gallagher's Gaussian 101-me Peaks Function and Gallagher's Gaussian 21-hi Peaks Function from Finck et al. (2010a), as well as the Ackley, Dixon-Price, Griewank and Levy functions from Jamil & Yang (2013).

First, we examine how different meta-objectives shape algorithm behavior by training three LQD variants (Section 3.2.2). Figure 3 shows the Pareto front of fitness versus novelty scores achieved by each algorithm on the training tasks. The results demonstrate that our framework successfully discovers state-of-the-art specialized algorithms: LQD (Q) outperforms the standard GA in fitness optimization ($p < 10^{-10}$), LQD (D) achieves superior novelty scores compared to Novelty Search ($p < 10^{-9}$), and LQD (QD) achieves Pareto dominance over both ME and DNS ($p < 10^{-3}$).

In addition to performance on meta-training tasks, our LQD framework demonstrates strong generalization to out-of-distribution (OOD) problems (Section B.2) and robust scaling properties (Section B.6). LQD (D) generally achieves superior novelty scores on unseen tasks, while LQD (Q) maintains competitive fitness. Furthermore, LQD shows effective scaling to population sizes and problem dimensions significantly larger than those encountered during training. A detailed analysis of these generalization capabilities, including performance across various OOD functions (see Figure 9) and an examination of scaling behavior (**??**), is presented in Appendix Section B.6.

### 4.3 ROBOT CONTROL TASKS

To evaluate generalization capabilities beyond abstract benchmarks, we evaluate LQD on a suite of challenging *out-of-distribution robot* control tasks. These experiments represent a significant departure from the meta-training conditions, featuring much higher-dimensional search spaces (up to 584 parameters for the Ant robot), domain-specific behavioral descriptors based on foot contact patterns and velocities (rather than random projections), and fundamentally different fitness landscapes. We evaluate performance across Hopper, Walker2d, Half Cheetah, Ant and Arm robots with various descriptor configurations with a budget of 250,000 evaluations. In Section E.3, we give more details about robot control tasks.

LQD demonstrates remarkable generalization to this challenging domain (Figure 4). LQD (Q) and LQD (D) outperforms all baselines ($p < 0.005$) while LQD (QD) outperforms all baselines except DNS where it matchs its performance. These results are particularly notable given that LQD was never exposed to robotic control problems, high-dimensional search spaces, or domain-specific descriptors during training, suggesting the learned competition rules capture fundamental principles that generalize well beyond their training distribution. Additional results are provided in Section B.5.

# 5 ANALYSIS OF DISCOVERED LQD

After demonstrating LQD's effectiveness across various optimization tasks, we now investigate the key mechanisms behind its performance. We examine two key aspects: the emergence of population diversity even when optimizing solely for fitness (Section 5.1) and the local competition strategies discovered for each variant (Section 5.2).

## 5.1 EMERGENT DIVERSITY

A striking finding from our analysis is that even LQD (Q) variant trained purely for fitness optimization naturally maintain significant population diversity. As shown in Figure 5, while LQD (Q) outperforms GA in maximizing fitness, it simultaneously achieves substantially higher novelty scores despite never being trained for diversity. For example, on Walker2d, LQD (Q) consistently achieves novelty scores approximately 40% of ME's final diversity, while standard GA maintains less than 10% of ME's diversity, demonstrating much stronger convergence.

This emergent diversity in LQD (Q) suggests that meta-optimization has rediscovered a fundamental principle of evolutionary systems: maintaining a diverse population creates crucial stepping stones that enable the discovery of high-performing solutions. While traditional GAs

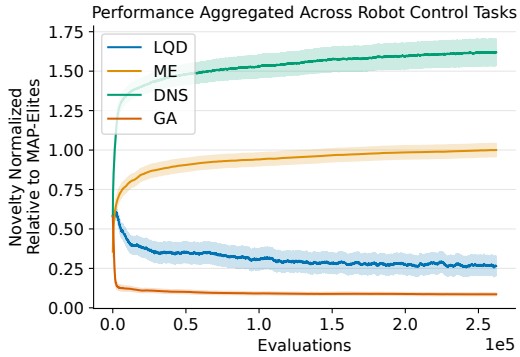

Figure 5: Novelty for LQD (Q) and baselines across robot control tasks. Lines show mean performance across 32 independent runs, with shaded regions indicating 95% confidence intervals.

often converge prematurely due to their global selection, LQD (Q) appears to have learned more sophisticated competition dynamics that naturally preserve promising intermediate solutions.

## 5.2 DISCOVERED LOCAL COMPETITION STRATEGIES

To understand how different meta-objectives shape the competition rules learned by LQD, we visualize the competition fitness values $\tilde{f}$ assigned by each variant across the descriptor space. Figure 6 shows these learned competition landscapes, where heatmaps represent the competition fitness that would be assigned to a solution with median raw fitness at each point in the descriptor space, based on the current state of the population (represented as colored dots).

LQD (D) develops a simple distance-based competition mechanism that rewards solutions for being far from existing ones, effectively rediscovering the core principle of novelty search.

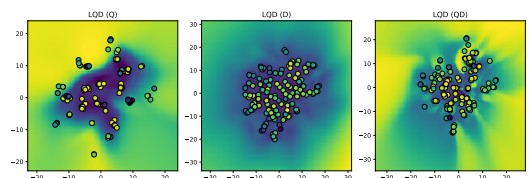

Figure 6: Visualization of learned competition landscapes across LQD variants. Points show evolved populations colored by raw fitness, while heatmaps represent the competition fitness $\tilde{f}$ that would be assigned to a solution with median fitness at each location. the axes represent the two dimensions of the descriptor space.

LQD (Q), despite being trained purely for fitness, learns a more nuanced strategy that creates fitness-sensitive reward patterns around promising solutions, explaining the emergent diversity observed in Section 5.1. LQD (QD) combines these approaches, creating directed exploration through competition rules that balance novelty seeking with fitness sensitivity.

These visualizations reveal that meta-optimization discovers qualitatively different competition strategies based on the meta-objective, from pure novelty-seeking to sophisticated hybrid mechanisms.

## 5.3 IMPACT OF DESCRIPTORS ON LQD

To understand if LQD leverages descriptor information to guide optimization, we conducted an ablation study comparing three configurations: standard LQD with task-specific descriptors, LQD

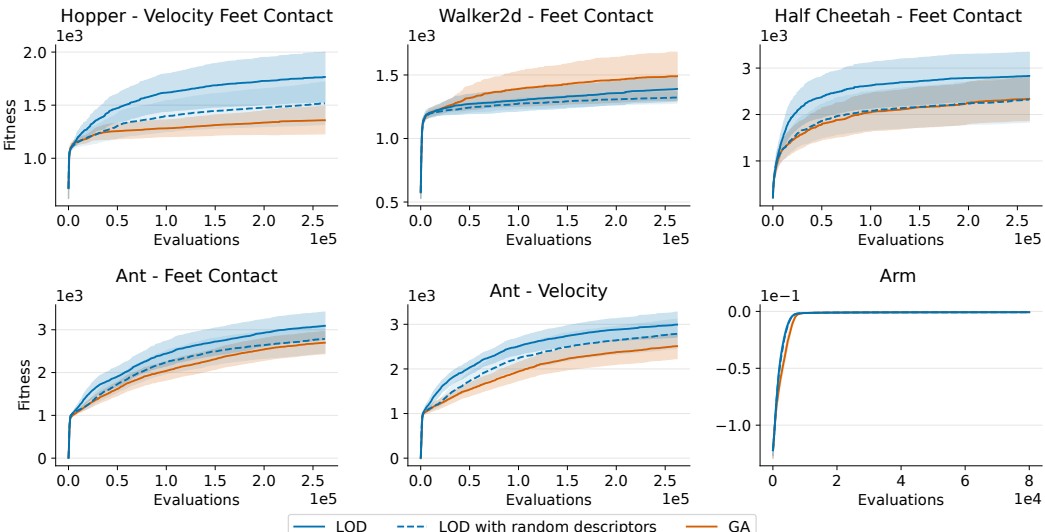

Figure 7: Impact of descriptors on optimization performance. Comparison of LQD (Q) with task-specific descriptors, LQD (Q) with random descriptors, and standard GA on robot control tasks. Lines show mean performance across 32 independent runs with 95% confidence intervals. The convergence of LQD (Q) with random descriptors to GA performance demonstrates that meaningful descriptor information is crucial for LQD's enhanced optimization capabilities.

with random descriptors (where descriptors are sampled from a standard normal distribution, breaking any relationship with the underlying solutions), and a standard GA. As shown in Figure 7, LQD with random descriptors performs similarly to GA across robot control tasks, suggesting that when descriptor information is meaningless, LQD essentially defaults to global competition. In contrast, LQD with task-specific descriptors significantly outperforms both alternatives, demonstrating that the learned competition rules effectively exploit the structure encoded in the descriptor space to identify promising stepping stones.

The algorithm appears to have discovered that descriptor-based similarity can indicate promising search directions, using this information to maintain strategic diversity that facilitates the discovery of high-performing solutions. This pattern holds across different robot morphologies and descriptor types, providing strong evidence that LQD's enhanced performance stems from its ability to leverage meaningful relationships encoded in the descriptor space.

# 6 CONCLUSION

Our work demonstrates that meta-learning can discover sophisticated Quality-Diversity algorithms that outperform traditional hand-designed approaches. Our Learned Quality-Diversity framework, which parameterizes competition rules using attention-based architectures, successfully discovers algorithms that capture complex relationships between solutions while maintaining the benefits of local competition that make biological evolution so powerful.

The discovered algorithms show several remarkable properties. First, they achieve superior or competitive performance compared to established baselines across various optimization tasks. Second, they demonstrate robust generalization, performing well on problems far outside their training distribution. Third, and perhaps most intriguingly, even when trained solely for fitness optimization, they naturally maintain significant population diversity, suggesting that meta-learning rediscovers diversity as an instrumental goal for achieving peak performance.

Our results also highlight areas for further improvement. While LQD variants excel at pure fitness maximization or novelty seeking, simultaneously optimizing both objectives remains challenging. LQD (QD) approaches but does not consistently outperform existing QD algorithms, especially on out-of-distribution domains, indicating that balancing quality and diversity remains a fundamental challenge in evolutionary computation.

ACKNOWLEDGMENTS

The authors would like to acknowledge the use of a large language model (LLM) to improve the grammar, clarity, and overall presentation of this manuscript. The authors reviewed, edited, and take full responsibility for the final content.

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

# A    RELATED WORK

## A.1    DIVERSITY IN GENETIC ALGORITHMS

Evolution's remarkable capacity for generating diverse, well-adapted species has inspired numerous algorithmic approaches in evolutionary computation. Early diversity maintenance techniques focused on preventing premature convergence through mechanisms like fitness sharing (Goldberg & Richardson, 1987) and hierarchical fair competition (Hu et al., 2005), which restrict competition to occur within species or niches. The NEAT algorithm (Stanley & Miikkulainen, 2002) demonstrated the power of this approach by using speciation to protect structural innovations in evolving neural networks, allowing novel architectures to optimize before competing with established solutions. Novelty Search (Lehman & Stanley, 2011a) took a radical step by completely abandoning the objective function, instead rewarding solutions solely for being different from their predecessors.

## A.2    QUALITY-DIVERSITY

Building on these insights, QD optimization has emerged as a distinct family of algorithms that combine novelty with local competition. Novelty Search with Local Competition (Lehman & Stanley, 2011b) pioneered this approach by having solutions compete only against behavioral neighbors, while MAP-Elites (Mouret & Clune, 2015) formalized it through a grid-based architecture where each cell maintains its highest-performing solution. These approaches have proven particularly valuable in robotics (Cully et al., 2015), where diverse behavioral repertoires enable adaptation. Despite relying on heuristics, recent theoretical analysis has confirmed that QD algorithms can provably outperform standard evolutionary strategies on NP-hard problems by avoiding local optima (Qian et al., 2024). Furthermore, the utility of diversity optimization extends beyond EC, having been successfully applied to Population-Based Reinforcement Learning to generate quality-similar diverse behaviors (Wu et al., 2023). However, current QD algorithms often rely on simple heuristics — grid-based approaches impose rigid discretization, while nearest-neighbor methods may miss important relationships between distant solutions. While recent advances like CMA-MAE (Fontaine & Nikolaidis, 2023) have significantly enhanced QD by integrating powerful variation operators (CMA-ES) to address search efficiency, our work focuses on the complementary challenge of the selection mechanism. We posit that meta-learning competition rules offers a distinct advantage when the primary bottleneck is not the optimizer, but the limitations of rigid archive structures.

## A.3    DISCOVERING ALGORITHMS VIA META-OPTIMIZATION

Over the course of machine learning's evolution, handcrafted components have gradually been replaced by modules learned directly from data. Recent work continues this trend by pursuing end-to-end discovery of gradient descent-based optimizers (Bengio et al., 1992; Andrychowicz et al., 2016; Metz et al., 2019), objective functions in Reinforcement Learning (Oh et al., 2020; Xu et al., 2020; Lu et al., 2022), the online tuning of hyperparameter schedules (Xu et al., 2018; Zahavy et al., 2020; Parker-Holder et al., 2022), and the meta-learning of complete learning algorithms (Wang et al., 2016; Kirsch & Schmidhuber, 2021; Kirsch et al., 2022). In these methods, appropriate inductive biases supplied by neural network parametrizations can guide and constrain the meta-search process. Here, our proposed LQD architecture leverages the equivariance properties encoded by the self-attention mechanism (Lee et al., 2019; Kossen et al., 2021; Tang & Ha, 2021), thereby defining a family of QD algorithms parameterized by neural networks.

## A.4    POPULATION-BASED OPTIMIZATION VIA META-OPTIMIZATION

The automation of evolutionary algorithm design has recently coalesced into the field of Meta-Black-Box-Optimization (Ma et al., 2025; Yang et al., 2025). Within this paradigm, several attempts have been made to automatically optimize evolutionary algorithms. For example, Chen et al. (2017); TV et al. (2019); Gomes et al. (2021) explored meta-learning entire algorithms for low-dimensional BBO using a sequence model to process solution candidates and/or their fitness values. Shala et al. (2020), on the other hand, introduced a meta-optimized policy to control the scalar search scale of CMA-ES (Hansen & Ostermeier, 2001). However, these methods often have limited generalization capabilities and are restricted to certain optimization domains, fixed population sizes, or fixed search

dimensions. Most closely related to our work are LES (Lange et al., 2023b) and LGA (Lange et al., 2023a). Both approaches relied on the invariance of dot-product self-attention (Lee et al., 2019; Tang & Ha, 2021) to generalize to unseen meta-training settings. Here, we successfully meta-optimize a parametrized Transformer model to discover various QD algorithms, which can outperform baseline algorithms on unseen optimization tasks.

# B ADDITIONAL RESULTS

## B.1 TRAINING BLACK-BOX OPTIMIZATION BENCHMARK TASKS

To provide a comprehensive performance overview on the training distribution, we compare each LQD variant and its corresponding baselines across all 22 meta-training BBOB tasks. Figure 8 presents the normalized scores for each specialized algorithm. These results reinforce our main findings, showing that the discovered LQD algorithms consistently achieve superior or competitive performance against established QD methods.

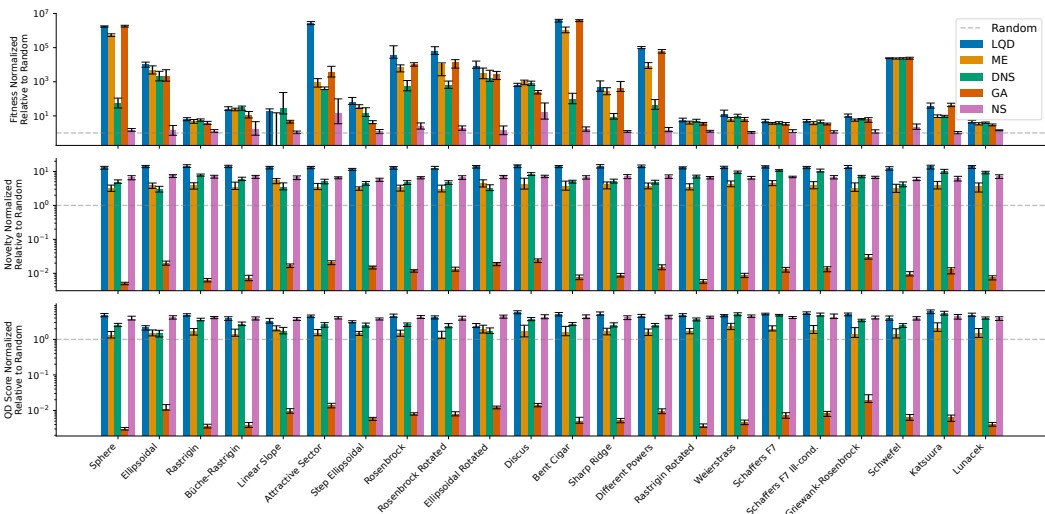

Figure 8: Performance comparison of the specialized LQD variants across the 22 training BBOB tasks. **Top:** The LQD (Q) variant evaluated on maximum fitness. **Middle:** The LQD (D) variant evaluated on mean novelty. **Bottom:** The LQD (QD) variant evaluated on the mean dominated novelty. All results are normalized relative to a random GA baseline (dashed line at y=1). Bars show median values across 32 runs, with error bars indicating the interquartile range. Higher values indicate better performance.

## B.2 EVALUATION BLACK-BOX OPTIMIZATION BENCHMARK TASKS

To assess generalization, we evaluate the discovered algorithms on six challenging out-of-distribution BBOB tasks not seen during meta-training. Figure 9 compares the performance of each specialized LQD variant and the baselines on these unseen functions. The results highlight the strong generalization capabilities of LQD, demonstrating that the learned competition rules remain effective well beyond the training distribution.

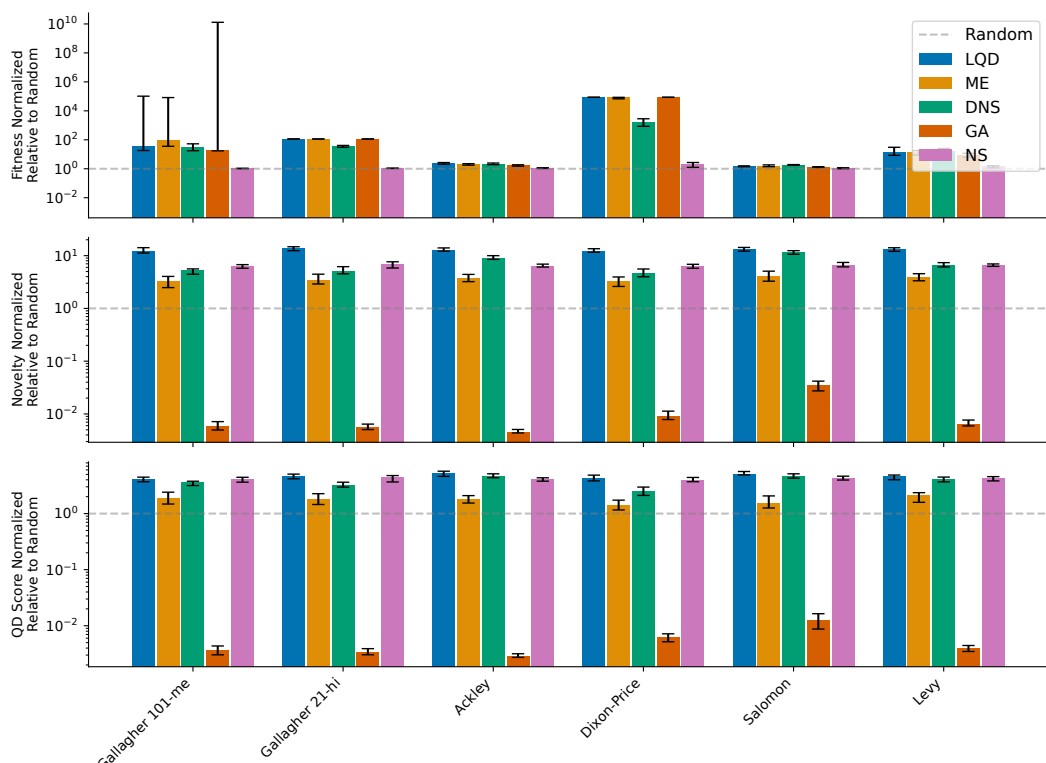

Figure 9: Performance comparison of the specialized LQD variants across the six out-of-distribution BBOB tasks. **Top:** The LQD (Q) variant evaluated on maximum fitness. **Middle:** The LQD (D) variant evaluated on mean novelty. **Bottom:** The LQD (QD) variant evaluated on the mean dominated novelty. All results are normalized relative to a random GA baseline (dashed line at y=1). Bars show median values across 32 runs, with error bars indicating the interquartile range. Higher values indicate better performance.

## B.3 PERFORMANCE ON BBOB TASKS

In this section, we provide detailed numerical results on the BBOB tasks to complement the aggregated bar charts. Tables 2 to 4 present the precise median scores obtained from our experiments, alongside the Interquartile Range (IQR) to illustrate performance variability across 32 independent runs. Overall, the results substantiate the superiority of the LQD variants. As detailed in Table 2, LQD (Q) consistently achieves the highest fitness on the vast majority of functions. Similarly, Table 3 demonstrates that LQD (D) is the strongest performer regarding exploration, significantly outperforming all baselines in mean novelty. Finally, Table 4 confirms that the LQD (QD) variant yields the best overall Quality-Diversity scores across most benchmark functions, underscoring its effectiveness in balancing local competition and global performance.

Table 2: **Maximum Fitness Analysis on BBOB Tasks.** The table reports the median maximum fitness reached after 1M evaluations, with the Interquartile Range (IQR) in parentheses calculated over 32 runs. Higher values indicate better performance. LQD (Q) achieves the best performance on the majority of functions, demonstrating superior pure optimization capabilities compared to standard baselines.

| Function | LQD (Q) | ME | DNS | GA | NS |
|---|---|---|---|---|---|
| Sphere | **-9.12** $(-9.88, -7.70) \cdot 10^{-5}$ | $-128.88 \ (-406.86, -49.10) \cdot 10^{-5}$ | $-6.03 \ (-9.47, -4.50)$ | $-4.51 \ (-87.73, -0.02) \cdot 10^{-2}$ | $-83.74 \ (-98.74, -71.21)$ |
| Ellipsoidal | **-4.66** $(-5.65, -2.84) \cdot 10^{2}$ | $-15.22 \ (-25.27, -8.90) \cdot 10^{2}$ | $-48.11 \ (-104.92, -24.04) \cdot 10^{2}$ | $-132.59 \ (-291.49, -41.00) \cdot 10^{2}$ | $-1.73 \ (-4.22, -1.41) \cdot 10^{6}$ |
| Rastrigin | **-1.08** $(-1.34, -0.89) \cdot 10^{2}$ | $-1.49 \ (-1.90, -1.19) \cdot 10^{2}$ | $-1.37 \ (-1.63, -1.14) \cdot 10^{2}$ | $-1.82 \ (-2.20, -1.47) \cdot 10^{2}$ | $-5.45 \ (-6.61, -4.27) \cdot 10^{2}$ |
| Büche-Rastrigin | $-1.97 \ (-2.46, -1.52) \cdot 10^{2}$ | $-2.32 \ (-2.83, -1.89) \cdot 10^{2}$ | **-1.82** $(-2.59, -1.54) \cdot 10^{2}$ | $-4.58 \ (-6.91, -2.88) \cdot 10^{2}$ | $-30.32 \ (-84.35, -12.98) \cdot 10^{2}$ |
| Linear Slope | $-4.77 \ (-11.05, 0.00)$ | **-0.00** $(-14.11, 0.00)$ | $-20.65 \ (-33.95, -12.40)$ | $-94.35 \ (-104.05, -76.71)$ | $-224.64 \ (-261.03, -195.72)$ |
| Attractive Sector | **-15.19** $(-23.92, -7.09)$ | $-43.44 \ (-61.54, -30.31)$ | $-73.17 \ (-85.70, -54.63)$ | $-77.21 \ (-99.90, -62.52)$ | $-1481.98 \ (-6067.63, -218.25)$ |
| Step Ellipsoidal | **-2.43** $(-3.86, -1.33) \cdot 10^{3}$ | $-3.57 \ (-5.20, -3.22) \cdot 10^{3}$ | $-10.20 \ (-15.45, -6.28) \cdot 10^{3}$ | $-29.91 \ (-34.76, -23.15) \cdot 10^{3}$ | $-95.73 \ (-134.64, -64.05) \cdot 10^{3}$ |
| Rosenbrock | **-8.49** $(-10.28, -4.05)$ | $-63.27 \ (-76.91, -15.26)$ | $-475.24 \ (-1024.31, -205.41)$ | $-60.62 \ (-127.78, -15.26)$ | $-4.65 \ (-7.61, -2.73) \cdot 10^{4}$ |
| Rosenbrock Rotated | **-10.03** $(-12.97, -8.54)$ | $-77.26 \ (-114.31, -21.78)$ | $-875.89 \ (-1393.01, -483.86)$ | $-138.21 \ (-219.63, -109.22)$ | $-9.53 \ (-14.97, -7.36) \cdot 10^{4}$ |
| Ellipsoidal Rotated | **-6.01** $(-7.75, -3.19) \cdot 10^{2}$ | $-28.10 \ (-46.25, -12.26) \cdot 10^{2}$ | $-61.85 \ (-136.59, -32.77) \cdot 10^{2}$ | $-132.21 \ (-236.49, -51.13) \cdot 10^{2}$ | $-2.52 \ (-4.21, -1.75) \cdot 10^{6}$ |
| Discus | $-50.06 \ (-57.23, -39.41)$ | **-27.57** $(-40.59, -21.45)$ | $-35.04 \ (-44.84, -27.18)$ | $-113.09 \ (-137.71, -91.97)$ | $-1448.06 \ (-3830.80, -371.99)$ |
| Bent Cigar | **-77.28** $(-93.59, -65.45)$ | $-2422.67 \ (-15858.92, -599.03)$ | $-9.53 \ (-11.52, -4.64) \cdot 10^{6}$ | $-8.98 \ (-18.53, -0.41) \cdot 10^{5}$ | $-1.35 \ (-2.26, -1.05) \cdot 10^{8}$ |
| Sharp Ridge | **-5.04** $(-8.74, -2.39)$ | $-16.87 \ (-42.68, -7.51)$ | $-408.29 \ (-489.99, -306.24)$ | $-138.76 \ (-225.23, -81.86)$ | $-1.67 \ (-1.94, -1.50) \cdot 10^{3}$ |
| Different Powers | **-1.09** $(-1.42, -0.86) \cdot 10^{-3}$ | $-72.51 \ (-128.53, -32.01) \cdot 10^{-3}$ | $-2.91 \ (-4.35, -1.69)$ | $-1.85 \ (-3.14, -0.67)$ | $-34.58 \ (-46.05, -26.16)$ |
| Rastrigin Rotated | **-1.08** $(-1.43, -0.88) \cdot 10^{2}$ | $-1.60 \ (-1.97, -1.33) \cdot 10^{2}$ | $-1.38 \ (-1.65, -1.15) \cdot 10^{2}$ | $-1.82 \ (-2.25, -1.58) \cdot 10^{2}$ | $-5.43 \ (-7.11, -4.31) \cdot 10^{2}$ |
| Weierstrass | **-2.65** $(-3.67, -1.64)$ | $-5.79 \ (-7.45, -4.67)$ | $-4.33 \ (-5.14, -3.50)$ | $-5.44 \ (-7.65, -4.62)$ | $-32.88 \ (-37.09, -29.16)$ |
| Schaffers F7 | **-3.08** $(-3.94, -2.52)$ | $-4.33 \ (-4.99, -4.05)$ | $-4.52 \ (-4.84, -3.55)$ | $-4.62 \ (-5.48, -3.93)$ | $-12.79 \ (-15.61, -11.60)$ |
| Schaffers F7 Ill-cond. | **-10.78** $(-12.88, -9.11)$ | $-15.34 \ (-18.67, -12.88)$ | $-12.59 \ (-15.64, -10.40)$ | $-16.34 \ (-18.87, -14.85)$ | $-47.44 \ (-60.47, -40.16)$ |
| Griewank-Rosenbrock | **-3.25** $(-3.95, -2.61)$ | $-6.08 \ (-6.75, -4.97)$ | $-5.26 \ (-6.19, -4.90)$ | $-5.32 \ (-7.86, -4.14)$ | $-28.35 \ (-35.59, -21.90)$ |
| Schwefel | **-1.77** $(-1.91, -1.70)$ | $-1.93 \ (-2.09, -1.76)$ | $-2.19 \ (-2.39, -1.99)$ | $-1.84 \ (-2.07, -1.58)$ | $-1.62 \ (-3.25, -1.01) \cdot 10^{4}$ |
| Katsuura | $-0.12 \ (-0.14, -0.08)$ | $-0.54 \ (-0.57, -0.44)$ | $-0.58 \ (-0.65, -0.49)$ | **-0.10** $(-0.14, -0.09)$ | $-3.35 \ (-4.26, -2.82)$ |
| Lunacek | **-80.51** $(-90.87, -63.36)$ | $-109.87 \ (-123.67, -91.34)$ | $-95.94 \ (-103.26, -86.78)$ | $-119.31 \ (-133.64, -104.09)$ | $-262.21 \ (-297.71, -226.77)$ |
| Gallagher 101-me | $-2.12 \ (-4.16, -0.52)$ | **-1.69** $(-2.32, -0.13)$ | $-6.40 \ (-7.74, -2.36)$ | $-4.16 \ (-5.91, -1.95)$ | $-72.70 \ (-74.99, -67.73)$ |
| Gallagher 21-hi | **-0.69** $(-0.69, -0.69)$ | $-0.70 \ (-0.73, -0.69)$ | $-6.01 \ (-7.42, -3.82)$ | $-0.78 \ (-1.90, -0.69)$ | $-73.93 \ (-77.76, -72.88)$ |
| Ackley | **-4.26** $(-4.83, -3.72)$ | $-4.85 \ (-5.66, -4.46)$ | $-4.93 \ (-5.38, -4.31)$ | $-5.74 \ (-6.57, -5.29)$ | $-9.06 \ (-9.82, -8.27)$ |
| Dixon-Price | **-0.67** $(-0.67, -0.67)$ | $-1.33 \ (-4.64, -0.85)$ | $-1.75 \ (-2.96, -0.81) \cdot 10^{2}$ | $-16.57 \ (-32.22, -7.66)$ | $-2.85 \ (-5.31, -1.98) \cdot 10^{4}$ |
| Griewank | $-5.91 \ (-25.77, -1.42) \cdot 10^{-2}$ | **-3.24** $(-5.07, -1.58) \cdot 10^{-2}$ | $-24.74 \ (-32.86, -21.36) \cdot 10^{-2}$ | $-5.57 \ (-17.06, -2.56) \cdot 10^{-2}$ | $-101.93 \ (-102.55, -101.17) \cdot 10^{-2}$ |
| Levy | $-2.73 \ (-4.07, -1.14)$ | **-2.65** $(-3.95, -2.19)$ | $-2.70 \ (-4.15, -1.52)$ | $-4.31 \ (-5.59, -2.95)$ | $-28.32 \ (-34.74, -21.55)$ |

Table 3: **Novelty mean Analysis on BBOB Tasks.** The table reports the median of the mean novelty, with the Interquartile Range (IQR) in parentheses. Higher values indicate a more diverse population covering a larger volume of the descriptor space. LQD (D) consistently outperforms all baselines, including specific novelty-seeking algorithms like NS and DNS, across all benchmark functions.

| Function | LQD (D) | ME | DNS | GA | NS |
|---|---|---|---|---|---|
| Sphere | **5.70** (4.99, 6.09) | 1.30 (1.08, 1.64) | 2.12 (1.87, 2.52) | $8.48$ (3.29, 10.10) $\cdot 10^{-3}$ | 2.74 (2.43, 3.20) |
| Ellipsoidal | **5.42** (4.85, 5.72) | 1.32 (1.17, 1.61) | 1.30 (1.18, 1.79) | $8.62$ (7.78, 9.49) $\cdot 10^{-3}$ | 2.72 (2.57, 3.06) |
| Rastrigin | **5.62** (5.15, 6.04) | 1.42 (1.13, 1.66) | 2.95 (2.71, 3.16) | $2.71$ (2.31, 3.11) $\cdot 10^{-3}$ | 2.83 (2.56, 3.15) |
| Büche-Rastrigin | **5.35** (5.06, 5.98) | 1.34 (1.11, 1.78) | 2.27 (1.98, 2.55) | $3.04$ (2.74, 3.95) $\cdot 10^{-3}$ | 2.84 (2.41, 3.25) |
| Linear Slope | **5.47** (4.98, 5.81) | 2.11 (1.80, 2.49) | 1.68 (1.37, 2.09) | $8.22$ (7.26, 9.16) $\cdot 10^{-3}$ | 2.78 (2.32, 3.01) |
| Attractive Sector | **5.79** (5.39, 6.24) | 1.47 (1.27, 1.81) | 2.31 (2.02, 2.62) | 0.01 (0.01, 0.01) | 2.90 (2.64, 3.14) |
| Step Ellipsoidal | **5.50** (5.17, 5.96) | 1.51 (1.30, 1.65) | 2.21 (2.04, 2.48) | $7.39$ (6.42, 7.84) $\cdot 10^{-3}$ | 2.66 (2.48, 2.90) |
| Rosenbrock | **5.48** (4.88, 5.81) | 1.33 (1.07, 1.60) | 2.04 (1.84, 2.14) | $6.90$ (5.92, 8.53) $\cdot 10^{-3}$ | 2.72 (2.51, 2.96) |
| Rosenbrock Rotated | **5.42** (5.00, 5.85) | 1.21 (1.03, 1.63) | 1.94 (1.81, 2.24) | $9.04$ (7.32, 10.25) $\cdot 10^{-3}$ | 2.69 (2.48, 3.11) |
| Ellipsoidal Rotated | **5.55** (5.12, 6.20) | 1.74 (1.31, 2.19) | 1.66 (1.20, 2.01) | $8.83$ (7.75, 10.43) $\cdot 10^{-3}$ | 2.85 (2.61, 3.05) |
| Discus | **5.79** (5.00, 6.15) | 1.56 (1.17, 2.32) | 3.15 (2.90, 3.34) | $9.23$ (8.13, 10.06) $\cdot 10^{-3}$ | 2.76 (2.52, 3.03) |
| Bent Cigar | **5.71** (5.10, 6.23) | 1.54 (1.18, 1.87) | 2.19 (1.91, 2.44) | $9.76$ (8.17, 10.99) $\cdot 10^{-3}$ | 2.80 (2.45, 3.29) |
| Sharp Ridge | **5.57** (4.92, 6.25) | 1.49 (1.19, 1.76) | 2.09 (1.82, 2.42) | $9.15$ (7.83, 10.72) $\cdot 10^{-3}$ | 2.70 (2.45, 3.07) |
| Different Powers | **5.46** (4.82, 5.73) | 1.34 (1.13, 1.52) | 1.89 (1.66, 2.14) | $9.55$ (8.35, 11.01) $\cdot 10^{-3}$ | 2.69 (2.32, 2.94) |
| Rastrigin Rotated | **5.65** (5.13, 6.06) | 1.40 (1.23, 1.70) | 2.99 (2.72, 3.22) | $2.74$ (2.41, 2.98) $\cdot 10^{-3}$ | 2.79 (2.69, 3.04) |
| Weierstrass | **5.36** (4.90, 5.90) | 1.63 (1.37, 1.91) | 3.73 (3.33, 4.06) | $3.78$ (3.39, 4.56) $\cdot 10^{-3}$ | 2.63 (2.44, 2.87) |
| Schaffers F7 | **5.72** (5.31, 6.17) | 1.72 (1.51, 2.20) | 4.41 (4.05, 4.64) | $5.39$ (4.83, 6.42) $\cdot 10^{-3}$ | 2.89 (2.70, 3.14) |
| Schaffers F7 Ill-cond. | **5.35** (4.81, 5.62) | 1.44 (1.17, 1.89) | 3.98 (3.67, 4.45) | $5.36$ (4.78, 6.60) $\cdot 10^{-3}$ | 2.63 (2.42, 2.97) |
| Griewank-Rosenbrock | **5.57** (5.22, 6.15) | 1.27 (1.08, 1.90) | 2.88 (2.68, 3.16) | 0.01 (0.01, 0.02) | 2.79 (2.58, 3.09) |
| Schwefel | **5.49** (5.05, 6.34) | 1.36 (1.11, 1.72) | 1.99 (1.75, 2.31) | $5.19$ (4.71, 5.83) $\cdot 10^{-3}$ | 2.79 (2.52, 3.07) |
| Katsuura | **5.65** (4.82, 6.39) | 1.61 (1.27, 2.03) | 4.00 (3.58, 4.60) | $5.58$ (4.23, 7.02) $\cdot 10^{-3}$ | 2.63 (2.22, 3.00) |
| Lunacek | **5.69** (5.11, 6.30) | 1.42 (1.03, 1.67) | 3.87 (3.45, 4.11) | $3.32$ (2.94, 3.84) $\cdot 10^{-3}$ | 2.89 (2.57, 3.19) |
| Gallagher 101-me | **5.53** (5.01, 6.31) | 1.43 (1.06, 1.76) | 2.33 (2.18, 2.65) | 0.01 (0.01, 0.01) | 2.75 (2.60, 3.33) |
| Gallagher 21-hi | **5.82** (5.18, 6.17) | 1.43 (1.15, 1.73) | 2.35 (2.09, 2.76) | $8.88$ (6.08, 9.95) $\cdot 10^{-3}$ | 2.92 (2.54, 3.18) |
| Ackley | **5.43** (5.14, 5.79) | 1.42 (1.29, 1.68) | 3.56 (3.32, 3.85) | $2.12$ (2.01, 2.22) $\cdot 10^{-3}$ | 2.75 (2.56, 2.95) |
| Dixon-Price | **5.28** (5.01, 5.80) | 1.35 (1.15, 1.60) | 1.99 (1.75, 2.53) | $9.82$ (8.70, 11.67) $\cdot 10^{-3}$ | 2.66 (2.51, 2.96) |
| Griewank | **5.36** (4.96, 6.21) | 1.31 (1.06, 1.77) | 2.26 (2.11, 2.54) | $8.06$ (5.64, 10.02) $\cdot 10^{-3}$ | 2.72 (2.42, 3.12) |
| Levy | **5.70** (5.32, 6.13) | 1.60 (1.32, 1.87) | 2.89 (2.62, 3.18) | $4.04$ (3.34, 4.89) $\cdot 10^{-3}$ | 2.89 (2.75, 3.08) |

Table 4: **QD Score Analysis on BBOB Tasks.** The table reports the median of the mean dominated novelty—a metric balancing fitness and diversity—with the Interquartile Range (IQR) in parentheses. Higher values indicate a population that effectively maintains high-performing solutions across diverse niches. LQD (QD) achieves the highest scores on the majority of tasks, successfully balancing exploration and exploitation.

| Function | LQD (QD) | ME | DNS | GA | NS |
|---|---|---|---|---|---|
| Sphere | **5.75** (5.14, 6.51) | 1.57 (1.31, 1.92) | 3.03 (2.65, 3.58) | 0.02 (0.01, 0.02) | 4.58 (4.07, 5.43) |
| Ellipsoidal | 2.97 (2.24, 3.56) | 1.60 (1.42, 1.94) | 1.91 (1.75, 2.64) | 0.02 (0.01, 0.02) | **4.64** (4.32, 5.26) |
| Rastrigin | **5.71** (5.34, 6.04) | 1.89 (1.51, 2.15) | 4.08 (3.81, 4.44) | $4.68$ (3.99, 5.34) $\cdot 10^{-3}$ | 4.83 (4.33, 5.26) |
| Büche-Rastrigin | 4.82 (4.19, 5.16) | 1.72 (1.46, 2.29) | 3.23 (2.90, 3.64) | $5.12$ (4.46, 6.67) $\cdot 10^{-3}$ | **4.89** (4.25, 5.12) |
| Linear Slope | 4.81 (4.19, 5.34) | 2.58 (2.21, 3.17) | 2.61 (2.13, 3.25) | 0.02 (0.01, 0.02) | **5.16** (4.57, 5.59) |
| Attractive Sector | **5.49** (4.95, 6.16) | 1.76 (1.50, 2.18) | 3.40 (2.92, 3.71) | 0.02 (0.02, 0.02) | 5.09 (4.58, 5.47) |
| Step Ellipsoidal | 4.38 (3.89, 4.83) | 1.81 (1.58, 1.98) | 3.30 (2.97, 3.66) | $7.39$ (6.42, 7.84) $\cdot 10^{-3}$ | **4.65** (4.18, 4.87) |
| Rosenbrock | **5.21** (4.47, 5.58) | 1.60 (1.30, 1.90) | 2.89 (2.67, 3.07) | 0.01 (0.01, 0.02) | 4.55 (4.30, 5.10) |
| Rosenbrock Rotated | **4.81** (4.48, 5.34) | 1.40 (1.23, 1.92) | 2.73 (2.57, 3.28) | 0.02 (0.01, 0.02) | 4.54 (4.17, 5.12) |
| Ellipsoidal Rotated | 3.20 (2.91, 3.51) | 2.07 (1.55, 2.65) | 2.38 (1.74, 2.98) | 0.02 (0.01, 0.02) | **4.98** (4.43, 5.34) |
| Discus | **6.99** (6.10, 7.40) | 1.89 (1.39, 2.84) | 4.28 (4.05, 4.49) | 0.02 (0.01, 0.02) | 5.04 (4.61, 5.60) |
| Bent Cigar | **5.64** (4.96, 6.20) | 1.81 (1.41, 2.21) | 3.16 (2.70, 3.53) | 0.02 (0.01, 0.02) | 4.83 (4.22, 5.60) |
| Sharp Ridge | **5.89** (5.16, 6.65) | 1.77 (1.42, 2.06) | 3.01 (2.60, 3.44) | 0.02 (0.01, 0.02) | 4.57 (4.18, 5.37) |
| Different Powers | **4.82** (4.45, 5.64) | 1.59 (1.34, 1.82) | 2.73 (2.35, 3.09) | 0.02 (0.02, 0.02) | 4.70 (4.08, 5.15) |
| Rastrigin Rotated | **5.58** (5.12, 6.40) | 1.83 (1.59, 2.18) | 4.21 (3.82, 4.54) | $4.59$ (4.17, 5.06) $\cdot 10^{-3}$ | 4.77 (4.56, 5.24) |
| Weierstrass | 6.02 (5.59, 6.38) | 3.01 (2.31, 3.57) | **6.69** (5.90, 7.03) | $6.43$ (5.82, 7.75) $\cdot 10^{-3}$ | 6.11 (5.54, 6.66) |
| Schaffers F7 | **6.74** (6.23, 6.99) | 2.50 (2.13, 3.05) | 6.15 (5.69, 6.34) | $9.24$ (8.55, 11.39) $\cdot 10^{-3}$ | 5.35 (5.09, 5.78) |
| Schaffers F7 Ill-cond. | **6.28** (5.60, 6.84) | 2.07 (1.64, 2.65) | 5.46 (5.09, 6.12) | $9.39$ (8.53, 11.51) $\cdot 10^{-3}$ | 5.04 (4.53, 5.60) |
| Griewank-Rosenbrock | **5.64** (5.38, 6.50) | 1.62 (1.31, 2.45) | 3.88 (3.61, 4.25) | 0.03 (0.02, 0.03) | 4.89 (4.37, 5.28) |
| Schwefel | **4.78** (4.47, 5.65) | 1.65 (1.32, 2.09) | 2.96 (2.64, 3.42) | $9.05$ (7.96, 9.81) $\cdot 10^{-3}$ | 4.71 (4.34, 5.31) |
| Katsuura | **8.40** (7.32, 8.95) | 2.85 (2.28, 3.97) | 7.32 (6.48, 8.11) | $9.89$ (7.50, 12.52) $\cdot 10^{-3}$ | 6.17 (5.37, 7.26) |
| Lunacek | **6.43** (5.73, 7.07) | 1.92 (1.48, 2.32) | 5.12 (4.74, 5.42) | $5.58$ (5.03, 6.51) $\cdot 10^{-3}$ | 4.99 (4.33, 5.46) |
| Gallagher 101-me | **4.93** (4.33, 5.65) | 2.26 (1.73, 2.85) | 4.26 (3.88, 4.74) | 0.02 (0.02, 0.02) | 4.88 (4.45, 5.68) |
| Gallagher 21-hi | 5.04 (4.64, 5.56) | 2.06 (1.65, 2.53) | 3.85 (3.55, 4.48) | 0.02 (0.01, 0.02) | **5.11** (4.38, 5.53) |
| Ackley | **5.79** (5.44, 6.45) | 1.84 (1.70, 2.19) | 5.07 (4.92, 5.55) | $3.55$ (3.40, 3.83) $\cdot 10^{-3}$ | 4.76 (4.39, 5.17) |
| Dixon-Price | **4.92** (4.44, 5.77) | 1.60 (1.33, 1.91) | 2.90 (2.48, 3.58) | 0.02 (0.02, 0.02) | 4.44 (4.28, 5.22) |
| Griewank | **5.24** (4.87, 5.76) | 1.82 (1.43, 2.67) | 4.05 (3.57, 4.40) | 0.01 (0.01, 0.02) | 4.56 (4.26, 5.35) |
| Levy | **5.75** (5.24, 6.37) | 2.55 (1.93, 2.77) | 5.08 (4.57, 5.50) | $6.86$ (5.49, 8.57) $\cdot 10^{-3}$ | 5.49 (5.13, 5.93) |

## B.4   PERFORMANCE TRAJECTORIES ON OUT-OF-DISTRIBUTION BBOB TASKS

To complement the final performance results shown in Section B.2, this section provides detailed performance trajectories for each specialized LQD variant on the out-of-distribution BBOB tasks. The following figures illustrate the convergence of each algorithm and its baselines over 250,000 evaluations, showing how performance on the relevant metric evolves throughout the optimization process.

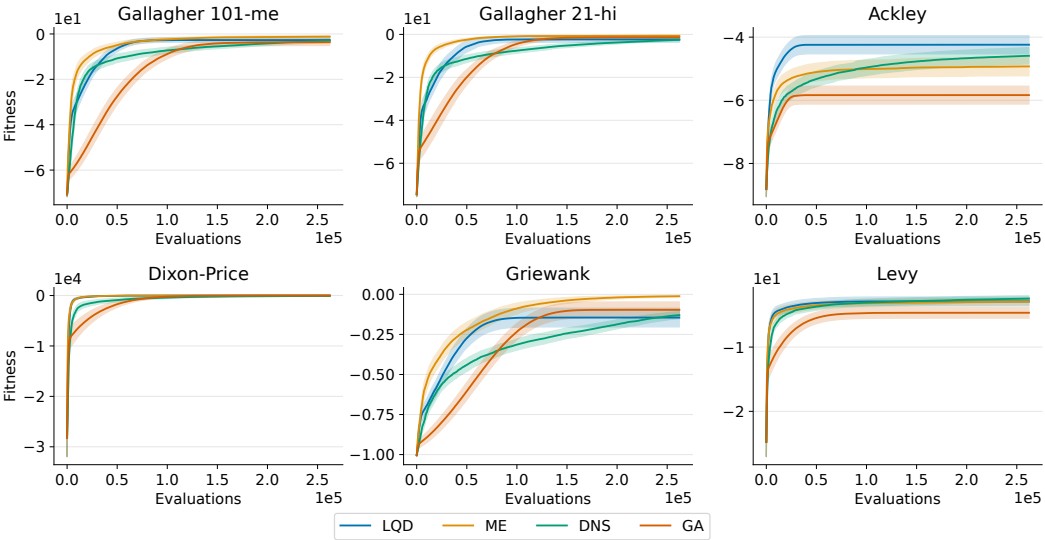

Figure 10: Performance trajectories for the LQD (Q) variant and baselines on the six out-of-distribution BBOB tasks. Each subplot shows the **maximum fitness** achieved as a function of the number of evaluations. Lines indicate the mean performance across 32 independent runs, while shaded regions represent the 95% confidence interval.

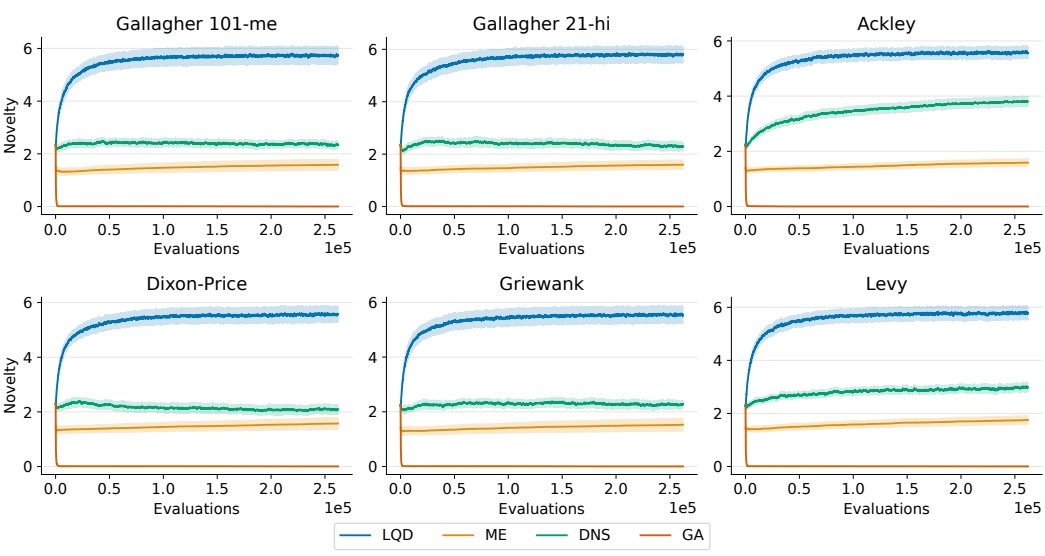

Figure 11: Performance trajectories for the LQD (D) variant and baselines on the six out-of-distribution BBOB tasks. Each subplot shows the **mean novelty** of the population as a function of the number of evaluations. Lines indicate the mean performance across 32 independent runs, while shaded regions represent the 95% confidence interval.

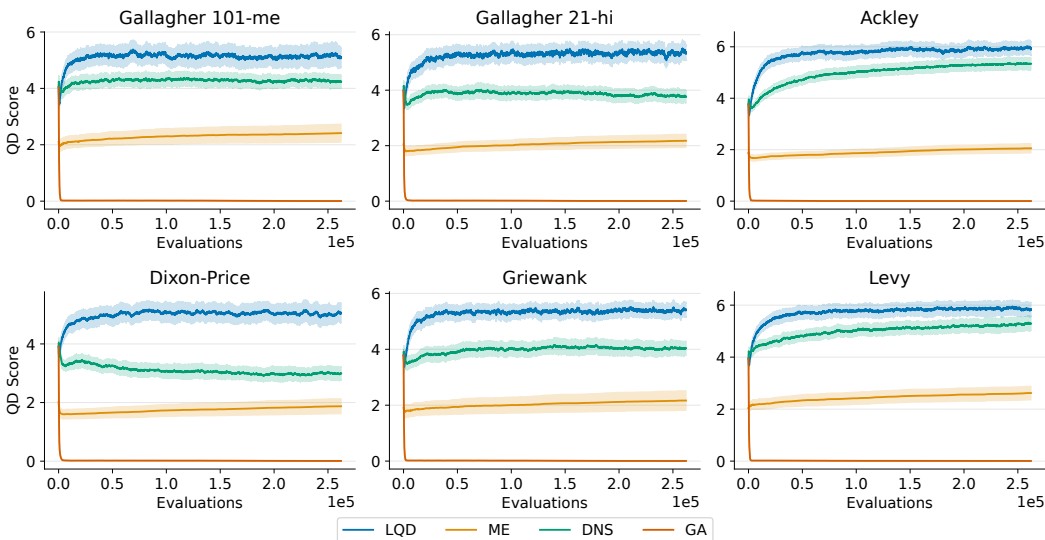

Figure 12: Performance trajectories for the LQD (QD) variant and baselines on the six out-of-distribution BBOB tasks. Each subplot shows the **mean dominated novelty** as a function of the number of evaluations. Lines indicate the mean performance across 32 independent runs, while shaded regions represent the 95% confidence interval.

## B.5 PERFORMANCE TRAJECTORIES ON ROBOT CONTROL TASKS

Similarly, this section details the performance trajectories of the LQD variants and baselines on the suite of robot control tasks. The figures show the learning curves for each algorithm, providing insight into their sample efficiency and convergence behavior in these high-dimensional, domain-specific environments.

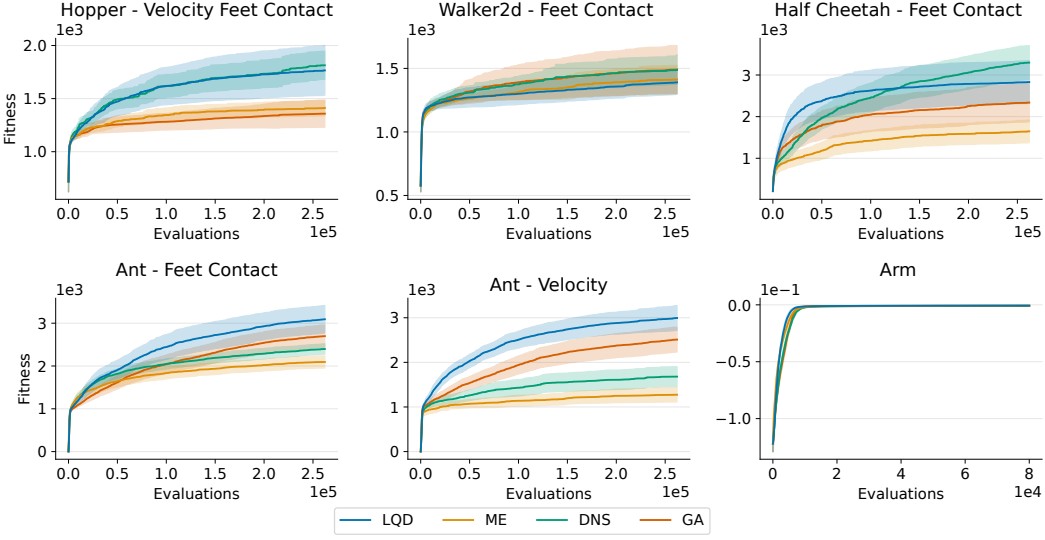

Figure 13: Performance trajectories for the LQD (Q) variant and baselines on the suite of robot control tasks. Each subplot shows the **maximum fitness** as a function of the number of evaluations. Lines indicate the mean performance across 32 independent runs, while shaded regions represent the 95% confidence interval.

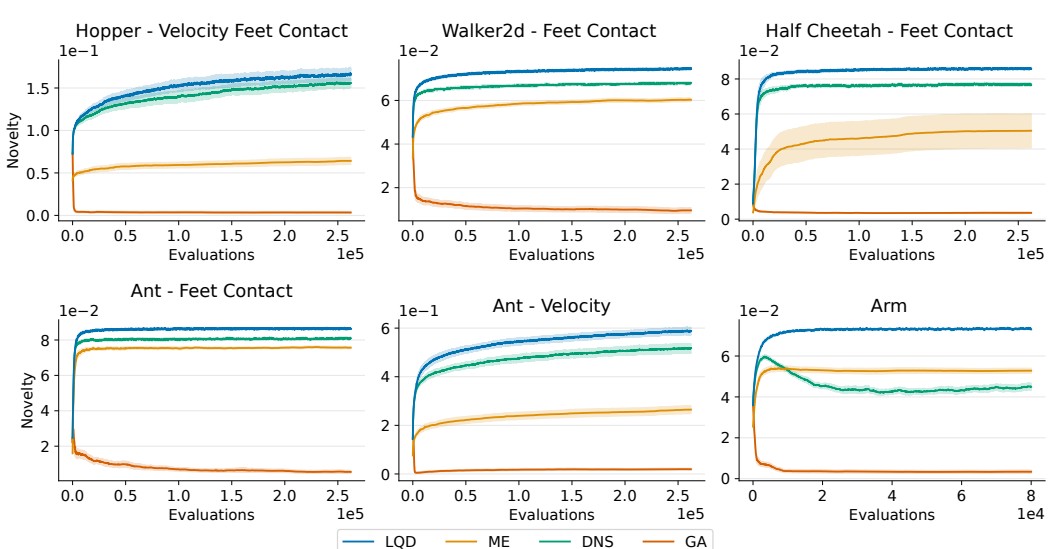

Figure 14: Performance trajectories for the LQD (D) variant and baselines on the suite of robot control tasks. Each subplot shows the **mean novelty** of the population as a function of the number of evaluations. Lines indicate the mean performance across 32 independent runs, while shaded regions represent the 95% confidence interval.

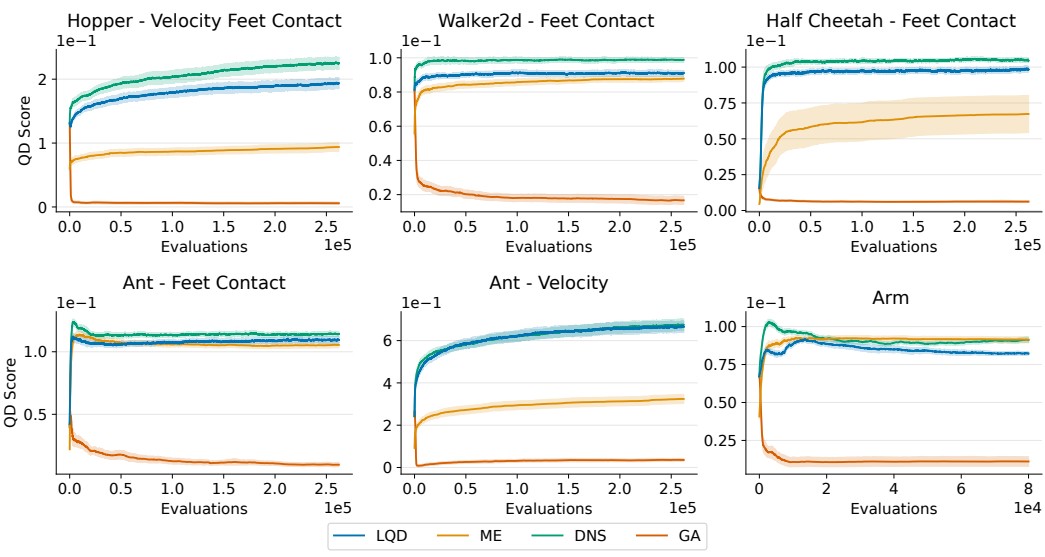

Figure 15: Performance trajectories for the LQD (QD) variant and baselines on the suite of robot control tasks. Each subplot shows the **mean dominated novelty** as a function of the number of evaluations. Lines indicate the mean performance across 32 independent runs, while shaded regions represent the 95% confidence interval.

### B.6 GENERALIZATION ON BLACK-BOX OPTIMIZATION BENCHMARK TASKS

Perhaps most impressively, LQD demonstrates robust scaling properties across different population sizes and problem dimensions, as shown in Figures 16 and 18. We evaluated the generalization capabilities of our learned algorithms on 6 BBOB functions, varying the problem dimension from 2 to 32 and the population size from 64 to 1024. This represents a significant departure from the meta-training distribution, where LQD only observed dimensions between 2 and 12 with a fixed population size of 128.

As illustrated in Figure 16, LQD (Q) outperforms MAP-Elites across the vast majority of configurations in terms of maximum fitness. Notably, the performance gap often widens as the dimensionality increases, suggesting that LQD (Q) scales more effectively to harder problems than the baseline. Regarding the diversity-oriented variants, Figure 17 and Figure 18 indicate that both LQD (D) and LQD (QD) maintain a performance advantage over MAP-Elites across most experimental settings. We observe a consistent trend where the performance gap widens as the problem dimensionality increases, further validating the method's ability to generalize to high-dimensional search spaces. However, this advantage tends to narrow as the population size grows larger. This suggests that while the competition rules learned on small populations ($N = 128$) remain effective, their relative dominance diminishes as the population increases significantly.

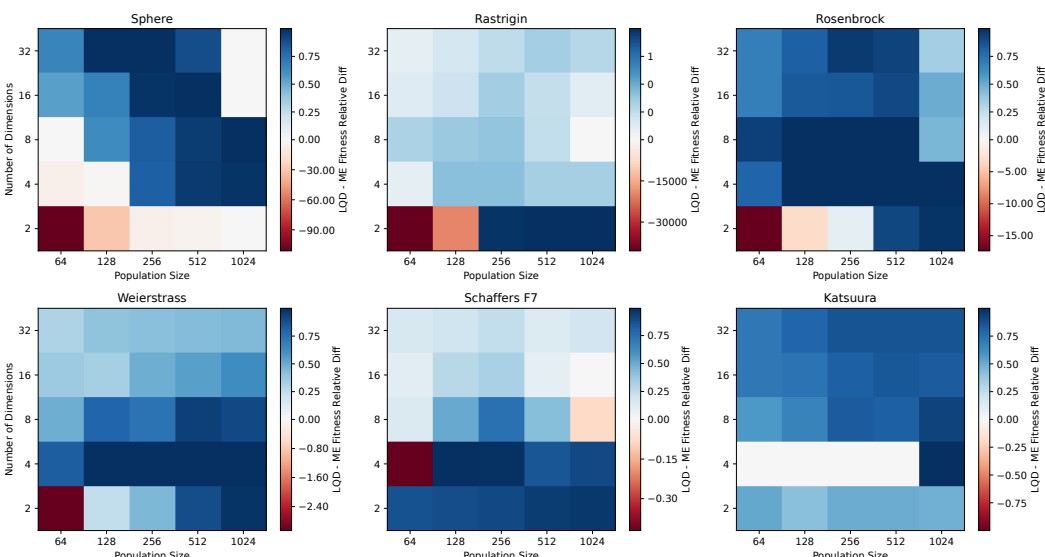

Figure 16: Generalization analysis of LQD (Q) across varying population sizes and search space dimensions. Heatmaps show the difference in **maximum fitness** between LQD (Q) and MAP-Elites (blue indicates LQD (Q) advantage, red indicates MAP-Elites advantage) across 6 BBOB tasks. LQD (Q) generalizes effectively to dimensions (up to 32) and population sizes (up to 1024) significantly larger than those seen during meta-training.

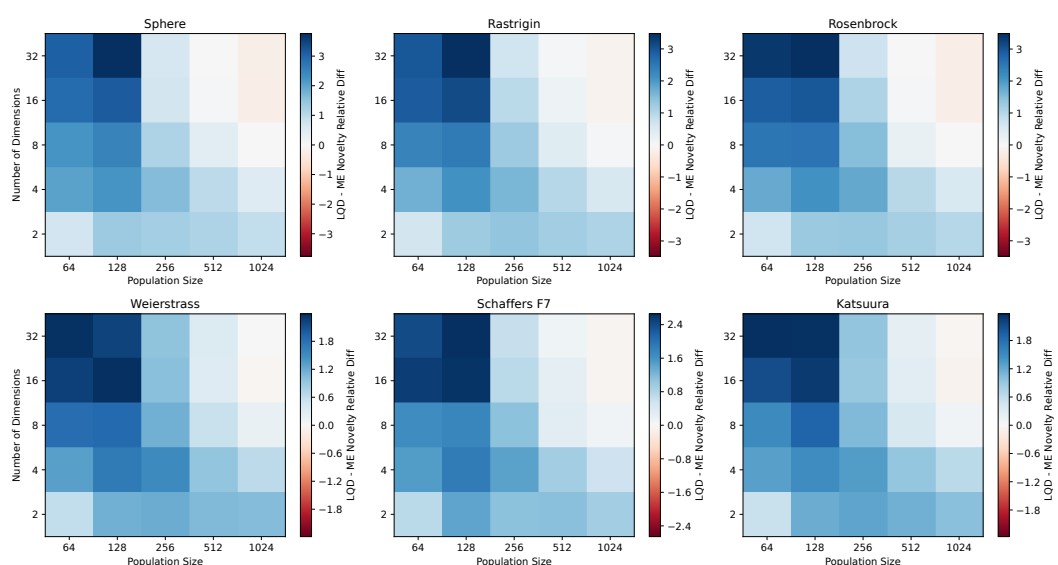

Figure 17: Generalization analysis of LQD (D) across varying population sizes and search space dimensions. Heatmaps show the difference in **mean novelty** between LQD (D) and MAP-Elites (blue indicates LQD (D) advantage, red indicates MAP-Elites advantage) across 6 BBOB tasks. LQD (D) generalizes effectively to dimensions (up to 32) and population sizes (up to 1024) significantly larger than those seen during meta-training.

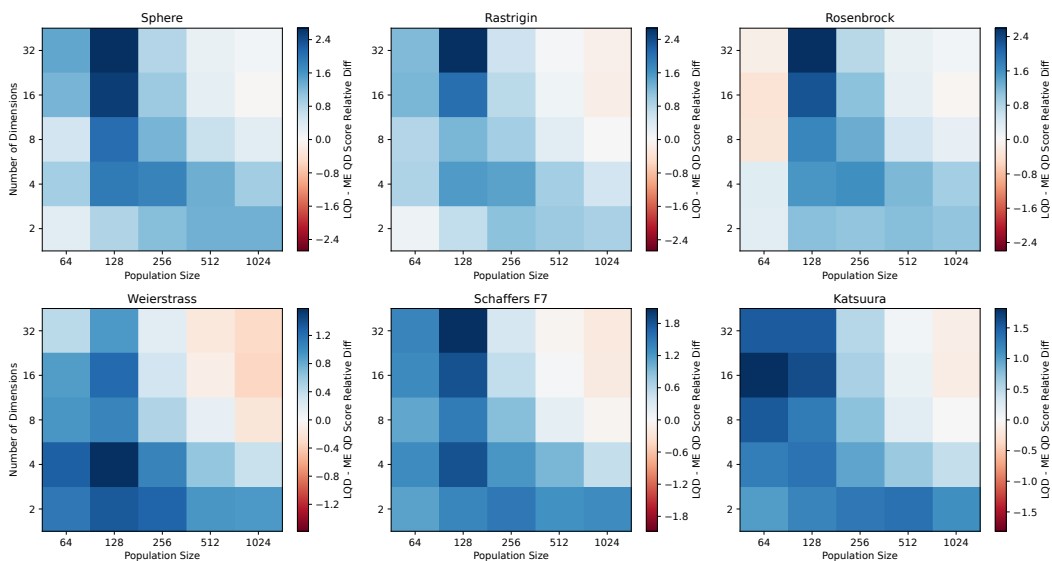

Figure 18: Generalization analysis of LQD (QD) across varying population sizes and search space dimensions. Heatmaps show the difference in **mean dominated novelty** between LQD (QD) and MAP-Elites (blue indicates LQD (QD) advantage, red indicates MAP-Elites advantage) across 6 BBOB tasks. LQD (QD) generalizes effectively to dimensions (up to 32) and population sizes (up to 1024) significantly larger than those seen during meta-training.

### B.7 POPULATIONS ON ROBOT CONTROL TASKS

We provide the populations obtained at the end of training for each algorithm on all robot control tasks in Figures 19 to 24. For each (algorithm, environment) pair, we select the most representative seed by picking the one with median score.

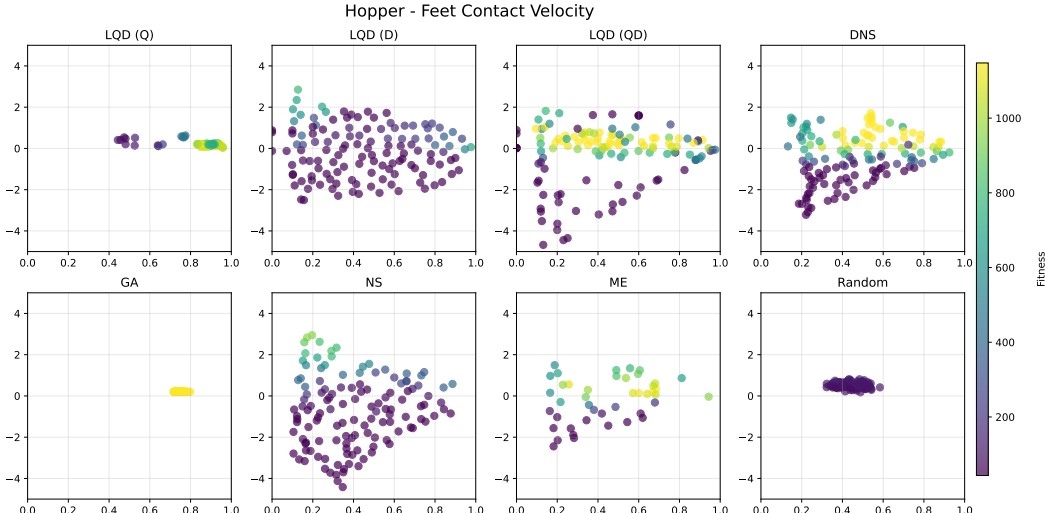

Figure 19: Populations for each algorithm on Hopper - Feet Contact Velocity. The axes represent the two dimensions of the descriptor space.

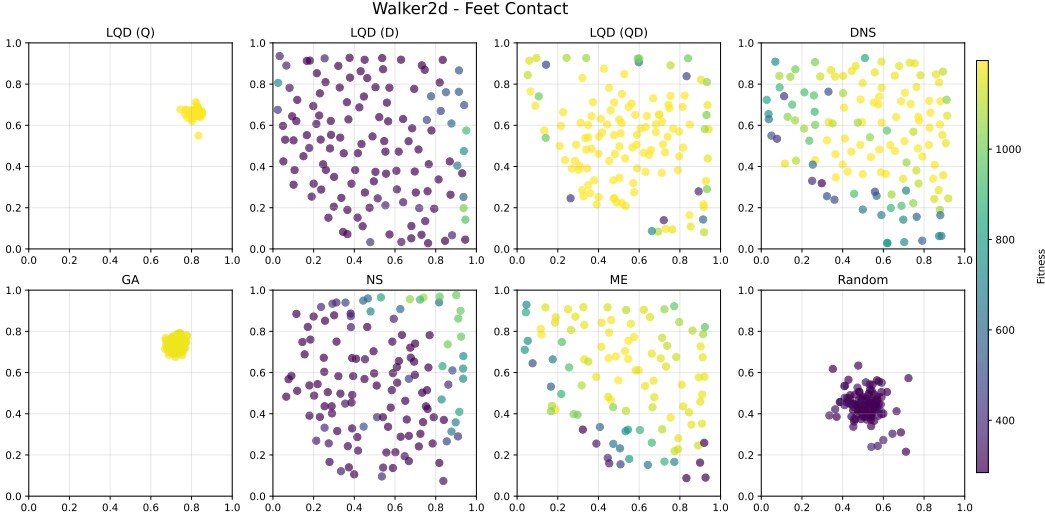

Figure 20: Populations for each algorithm on Walker2d - Feet Contact. The axes represent the two dimensions of the descriptor space.

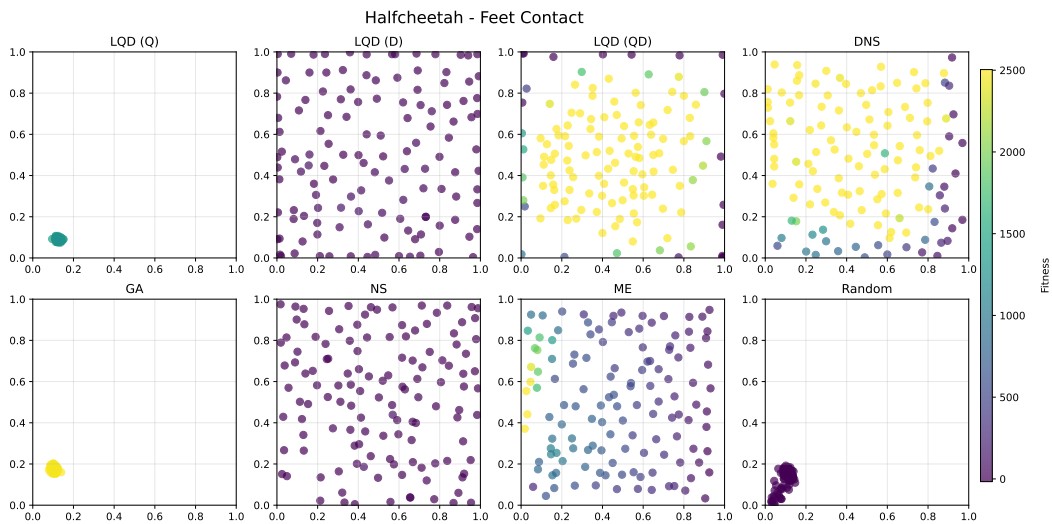

Figure 21: Populations for each algorithm on Halfcheetah - Feet Contact. The axes represent the two dimensions of the descriptor space.

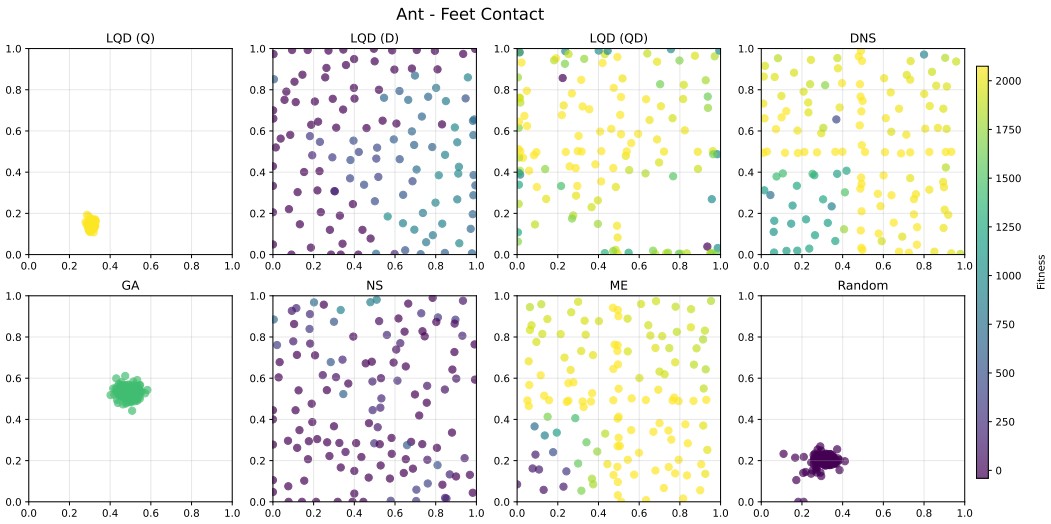

Figure 22: Populations for each algorithm on Ant - Feet Contact. The axes represent the two dimensions of the descriptor space.

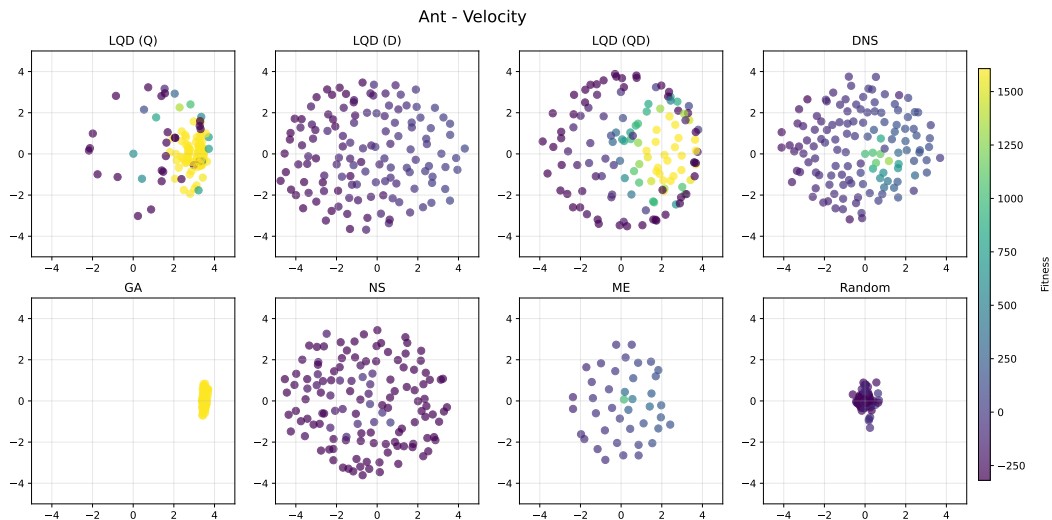

Figure 23: Populations for each algorithm on Ant - Velocity. The axes represent the two dimensions of the descriptor space.

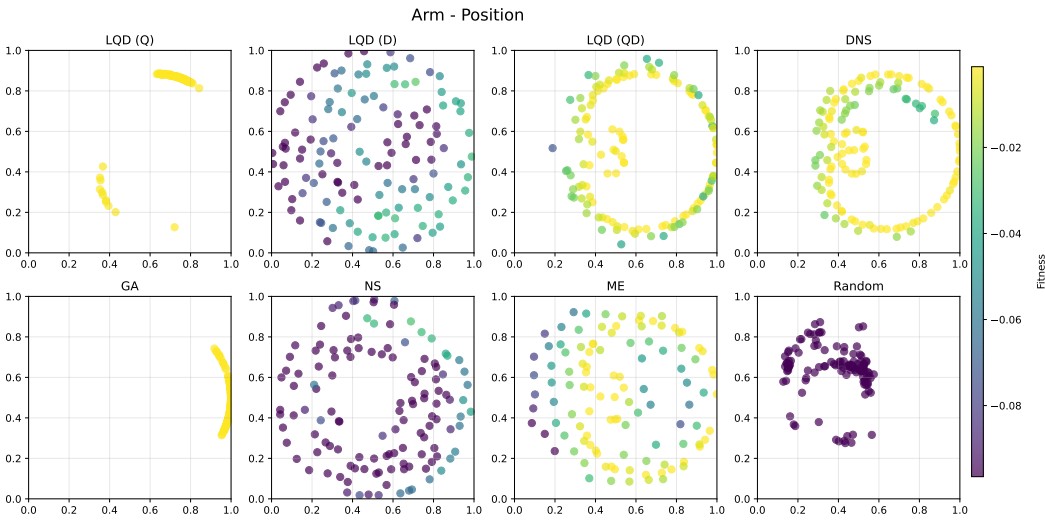

Figure 24: Populations for each algorithm on Arm - Position. The axes represent the two dimensions of the descriptor space.

# C  SET OPERATIONS VIA DOT-PRODUCT ATTENTION

Evolutionary algorithms fundamentally operate on populations — unordered sets of individuals. This set-based nature of populations creates a crucial requirement: any operation performed on the population must be invariant to the ordering of individuals.

Scaled dot-product attention (Vaswani et al., 2017) provides an elegant solution to this requirement through its inherent permutation equivariance properties. Consider a set of $N$ vectors represented as a matrix $\mathbf{X} \in \mathbb{R}^{N \times D}$. Scaled dot-product attention projects these elements into three distinct $D_K$-dimensional latent spaces — queries $\mathbf{Q}$, keys $\mathbf{K}$, and values $\mathbf{V}$. The output $\mathbf{Y}$ is then computed as a weighted combination of the values:

$$\mathbf{Q} = \mathbf{X}\mathbf{W}_Q \in \mathbb{R}^{N \times D_K}$$

$$\mathbf{K} = \mathbf{X}\mathbf{W}_K \in \mathbb{R}^{N \times D_K} \qquad \mathbf{Y} = \mathrm{softmax}\left(\frac{\mathbf{Q}\mathbf{K}^\top}{\sqrt{D_K}}\right)\mathbf{V} \in \mathbb{R}^{N \times D_K}$$

$$\mathbf{V} = \mathbf{X}\mathbf{W}_V \in \mathbb{R}^{N \times D_K}$$

Crucially, this transformation exhibits permutation equivariance: permuting the rows of the input population matrix $\mathbf{X}$ results in the same permutation being applied to the rows of the output $\mathbf{Y}$ (Vaswani et al., 2017). This property makes scaled dot-product attention and transformer architectures particularly well-suited for evolutionary algorithms, as it naturally preserves the set-based structure of populations while enabling complex interactions between individuals (Lange et al., 2023b;a; 2024).

## D  HYPERPARAMETERS

Our meta-optimization framework involves three sets of hyperparameters, detailed in Table 5. For the outer loop meta-optimization, we use SNES with an initial sigma $\sigma = 0.04$. The meta-population size ($M = 512$) and meta-batch size ($K = 512$) were chosen to provide stable gradient estimates while remaining computationally tractable. We run the meta-optimization for 16,384 generations to ensure convergence of the learned algorithms.

Table 5: Meta-black-box optimization hyperparameters

| Parameter | Value |
|---|---|
| Meta-ES | SNES |
| $\sigma$ init | 0.04 |
| Meta-population size $M$ | 512 |
| Meta-batch size $K$ | 512 |
| Num. meta-generations | 16,384 |
| $\alpha$ | 0.5 |
| Population size $N$ | 128 |
| Reproduction batch size $B$ | 32 |
| Num. generations $T$ | 256 |
| Num. layers | 6 |
| Num. features $D_K$ | 16 |
| Num. heads | 4 |

For the inner loop optimization, we use relatively modest population sizes ($N = 128$) and reproduction batch sizes ($B = 32$). Each inner loop runs for $T = 256$ generations, providing sufficient time for the algorithms to demonstrate their optimization capabilities while keeping meta-training computationally feasible.

The transformer architecture hyperparameters were selected to balance expressiveness and computational efficiency. We use 6 attention layers with 16 features per layer and 4 attention heads, resulting in 10,273 trainable parameters. This architecture provides sufficient capacity to learn sophisticated competition rules while remaining small enough to meta-optimize effectively.

# E  TASKS

## E.1  META-BLACK-BOX OPTIMIZATION TASKS

For meta-optimization, we utilize 22 functions from the Black-Box Optimization Benchmark (BBOB) suite (Finck et al., 2010a). For these mathematical optimization problems, which lack inherent physical behaviors, we define the descriptor space via random projection. Specifically, the high-dimensional genotype of each solution is projected onto a 2-dimensional descriptor space using a fixed, task-specific random matrix, as detailed in Section 3.2.1.    NEW

These functions were carefully selected to represent diverse optimization challenges across five categories: separable functions, low/moderate conditioning, high conditioning with unimodality, multimodal functions with adequate global structure, and multimodal functions with weak global structure. We exclude two computationally intensive functions (Gallagher's Gaussian 101-me Peaks Function (Finck et al., 2010a, p. 105) and 21-hi Peaks Function (Finck et al., 2010a, p. 110)) from the training set, reserving them instead for out-of-distribution evaluation. Table 6 provides a comprehensive overview of the training functions and their properties.

Table 6: Meta-black-box optimization tasks

| Function | Property | Reference |
|---|---|---|
| Sphere Function | Separable | Finck et al. (2010a, p. 5) |
| Ellipsoidal Function | Separable | Finck et al. (2010a, p. 10) |
| Rastrigin Function | Separable | Finck et al. (2010a, p. 15) |
| Büche-Rastrigin Function | Separable | Finck et al. (2010a, p. 20) |
| Linear Slope | Separable | Finck et al. (2010a, p. 25) |
| Attractive Sector Function | Low or moderate conditioning | Finck et al. (2010a, p. 30) |
| Step Ellipsoidal Function | Low or moderate conditioning | Finck et al. (2010a, p. 35) |
| Rosenbrock Function | Low or moderate conditioning | Finck et al. (2010a, p. 40) |
| Rosenbrock Function, rotated | Low or moderate conditioning | Finck et al. (2010a, p. 45) |
| Ellipsoidal Function, rotated | High conditioning and unimodal | Finck et al. (2010a, p. 50) |
| Discus Function | High conditioning and unimodal | Finck et al. (2010a, p. 55) |
| Bent Cigar Function | High conditioning and unimodal | Finck et al. (2010a, p. 60) |
| Sharp Ridge Function | High conditioning and unimodal | Finck et al. (2010a, p. 65) |
| Different Powers Function | High conditioning and unimodal | Finck et al. (2010a, p. 70) |
| Rastrigin Function | Multimodal with adequate global structure | Finck et al. (2010a, p. 75) |
| Weierstrass Function | Multimodal with adequate global structure | Finck et al. (2010a, p. 80) |
| Schaffers F7 Function | Multimodal with adequate global structure | Finck et al. (2010a, p. 85) |
| Schaffers F7 Function, ill-conditioned | Multimodal with adequate global structure | Finck et al. (2010a, p. 90) |
| Griewank-Rosenbrock | Multimodal with adequate global structure | Finck et al. (2010a, p. 95) |
| Schwefel Function | Multimodal with weak global structure | Finck et al. (2010a, p. 100) |
| Katsuura Function | Multimodal with weak global structure | Finck et al. (2010a, p. 115) |
| Lunacek bi-Rastrigin Function | Multimodal with weak global structure | Finck et al. (2010a, p. 120) |

## E.2  BLACK-BOX OPTIMIZATION TASKS

To rigorously evaluate generalization, we test on six challenging out-of-distribution functions (Table 7). These include the two held-out BBOB functions and four additional benchmarks from Jamil & Yang (2013). The selected functions feature different characteristics from the training set, including the Ackley function (known for its narrow global optimum), the Dixon-Price function (which tests optimization in narrow valleys), and the Levy function (characterized by numerous local optima).

## E.3  ROBOT CONTROL TASKS

We evaluate our method on six robotic control tasks implemented in the Brax physics simulator (Freeman et al., 2021) and one arm reaching task from Cully et al. (2015). These tasks represent significant departures from the training distribution, featuring high-dimensional search spaces (up to

Table 7: Out-of-distribution black-box optimization tasks

| Function | Property | Reference |
|---|---|---|
| Gallagher's 101-me Peaks Function | Multimodal with weak global structure | Finck et al. (2010a, p. 105) |
| Gallagher's 21-hi Peaks Function | Multimodal with weak global structure | Finck et al. (2010a, p. 110) |
| Ackley Function | Non-Separable, multimodal | Jamil & Yang (2013, p. 5) |
| Dixon-Price Function | Non-Separable, unimodal | Jamil & Yang (2013, p. 15) |
| Salomon Function | Non-Separable, multimodal | Jamil & Yang (2013, p. 27) |
| Levy Function | Non-Separable, multimodal | Jamil & Yang (2013, p. 40) |

584 parameters), domain-specific behavioral descriptors, and complex fitness landscapes shaped by physics-based constraints. As detailed in Table 8, each environment presents unique challenges with specific descriptor definitions:

- **Locomotion Tasks (Hopper, Walker, Half Cheetah, and Ant):** The objective is to maximize forward velocity. We employ two types of behavioral descriptors:
  - *Feet Contact:* A binary vector aggregated over the episode representing the proportion of time each foot is in contact with the ground. This captures the gait pattern of the robot.
  - *Velocity:* A vector containing the average velocity of the robot's center of mass, capturing the direction and speed of movement.
- **Arm Task:** The objective is to reach a target with a redundant planar arm.
  - *Descriptor:* The final $(x, y)$ Cartesian coordinates of the end-effector.
  - *Fitness:* The negative standard deviation of the joint angles, utilized to favor smooth configurations among the diverse reaching solutions.

These tasks test both the scalability of our method to high-dimensional problems and its ability to leverage meaningful, domain-specific behavioral descriptors.

Table 8: Robot Control Tasks

| Environment | Fitness | Descriptor | Genotype size | Reference |
|---|---|---|---|---|
| Hopper | Forward velocity | Velocity and feet contact | 243 | Freeman et al. (2021) |
| Walker | Forward velocity | Feet contact | 390 | Freeman et al. (2021) |
| Half Cheetah | Forward velocity | Feet contact | 390 | Freeman et al. (2021) |
| Ant | Forward velocity | Velocity | 584 | Freeman et al. (2021) |
| Ant | Forward velocity | Feet contact | 584 | Freeman et al. (2021) |
| Arm | Joint angles std | Final position of end effector | 8 | Cully et al. (2015) |

