# OpenReview forum: "Discovering Quality-Diversity Algorithms via Meta-Black-Box Optimization"
_ICLR.cc/2026/Conference — Submitted to ICLR 2026_

### Official Review · Reviewer_Ziuf · 2025-10-21

**Soundness:** 3
**Presentation:** 3
**Contribution:** 2
**Rating:** 2
**Confidence:** 4

**Summary:**

This paper introduces a new MetaBBO approach to learn generalizable QD algorithm. Based on Dominated Novelty Search (DNS) algorithm, the authors uses neuroevolution based MetaBBO paradigm to train an attention-based scoring function. Given the fitnesses and pre-defined descriptor vector of a population of solutions, the proposed scoring neural network uses attention mechanism to process these raw features and then obtain the scores of solution population. The scores are used in a DNS framework to promote QD optimization. The neural network is trained by SNES (an evolutionary strategy), where the fitness for the neural network is accumulated as its average (median) performance when being used for scoring in DNS on a meta-task set (synthetic BBO functions). After training, the authors found that the obtained optimal neural network individual presents ideal generalization ability, which is beyond problem dimension, population size, etc.

**Strengths:**

The learned scoring policy at least shows competitive performance compared with QD baselines.

**Weaknesses:**

1. Lack of respect for related works.

I am researching closely within learning-assisted evolutionary computation, hence familiar with the concept so-called Meta-BBO (or MetaBBO, never mind). Although the authors have reviewed  works of Lange et al, I have to say that the first time I heard Meta-BBO is from another two teams' surveys [1][2], where the works of Lange et al are also included as pioneering works. However, if you authors have read these surveys, you would know that there are many MetaBBO works now. However, you almost neglect them all (I do not know if you are intended to) and only reviewed works before 2022 or latest works of Lange et al. How could a reader know whether your paper is valuable or not considering that you omit comparison (at least review) recent related works? Let alone QD algorithms, which from my experience the team of Qian et al is also famous with significant works such as [3][4]. I urge the authors reconsider their attitude on scientific research before submitting the paper.

[1] https://ieeexplore.ieee.org/abstract/document/10993463

[2] https://www.sciencedirect.com/science/article/pii/S2210650224003766

[3] https://openreview.net/forum?id=bLmSMXbqXr

[4] https://dl.acm.org/doi/abs/10.24963/ijcai.2024/773

2. Lack of novelty.

If I understand correctly, the major contribution of this paper is proposal of learning scoring mechanism in EAs. However, learning scoring function or selection pressure through transformer is not novel at all, either in Lange et al's work [5] or end-to-end MetaBBO such as GLHF [6]. The authors did not discuss the technical novelty compared to these works, which hinders me from getting the real motivation and specific technical contribution of them to tailor LQD for QD algorithm. Besides, the trend in existing MetaBBO indicates that controlling a part of an algorithm (e.g., in this paper the selection pressure) is not flexible enough to discover new algorithm [7][8] (I read these two insightful papers recently), since the fixed algorithmic architecture presents limited refinement potential. I expect the perspective from the authors.

[5] https://dl.acm.org/doi/abs/10.1145/3583131.3590496

[6] https://proceedings.neurips.cc/paper_files/paper/2024/hash/19e9a88d91917775b34fdad447ed8908-Abstract-Conference.html

[7] https://openreview.net/forum?id=vLJcd43U7a

[8] https://arxiv.org/abs/2505.17866

3. Lack of comparative validation.

I quote a claim of the authors: "A striking finding from our analysis is that even LQD (Q) variant trained purely for fitness optimization naturally maintain significant population diversity.". I think this conlusion is very tricky. On the one hand, if so, why LQD (Q) is worse than LQD (QD)? Is this indicating that the proposed training procedure is not good enough so that it can not learn very effective LQD (Q)? On the other hand, if so, why the authors did not compare their models with existing MetaBBO approaches that use fitness optimization, what if these existing approaches have emerged QD principle by only-fitness rewarding? Overall, the comparison is very limited to tell true effectiveness of LQD. BTW: why robot control needs QD?

4. Technical issues.

(1) What is the dimension of the descriptor space (D) used in this paper? the authors provide the dimension of training problem instances (n = 2 ~ 12), and claim that " By projecting high-dimensional genotypes into a lower-dimensional descriptor space while maintaining their relative distances,", however, on the one hand, they do not provide tie value of D, if D > 2, how this claim to be correct when problem dimension is 2? On the other hand, why the relative distance could be maintained under the standard normal condition? prove it.

(2) what does the reproduction size B denotes? it is not mentioned in the paper?

(3) I can not agree with the authors the proposed complex neuroevolution training is efficient, especially in experimental setting the authors claim that they use 8 H100 GPUs for this training task, use such huge computational resources to achieve competitive results with naive GA and MAP-Elite can not really prove the value of this work.

To summarize, I do not think this paper (at its current version) deserves an acceptance. Substantial improvement need to be made to ensure the novelty, motivation, significance and scientific integrity before this paper could be accepted. I am wiling to engage discussion with the authors in the rebuttal phase and promise timely response.

**Questions:**

See weaknesses.

---

> ### Author Response · Authors · 2025-11-20
>
> We thank **Reviewer Ziuf** for their review and for highlighting relevant literature in the growing field of MetaBBO. We take your concerns regarding related work seriously and address them below.
>
> ---
>
> > Lack of respect for related works. [...] you almost neglect them all (I do not know if you are intended to) and only reviewed works before 2022 or latest works of Lange et al.
>
> We would like to apologise and assure you there was no intent to neglect related work. We cited Lange et al. extensively because our method builds directly upon their specific architecture (Attention-based source policies). We appreciate you providing the references to the recent surveys and the work of Qian et al. We will expand our Related Work section to include these citations ([1]-[4] from your review) to better contextualize our work within the broader and rapidly evolving MetaBBO landscape.
>
> > Lack of novelty. [...] learning scoring function or selection pressure through transformer is not novel at all...
>
> We respectfully disagree with the assessment regarding novelty. While the *tool* (transformers for meta-learning) has been explored in single-objective optimization (as in Lange et al.), the **application to Quality-Diversity (QD)** is fundamentally novel.
>
> Existing QD algorithms have relied for over a decade on human-designed heuristics: specifically, rigid grid discretizations (MAP-Elites) or distance-threshold archives (Novelty Search). These heuristics are the primary bottleneck preventing true open-endedness. Our work is the first to show that **meta-learning can discover a new type of QD archive mechanism** that breaks free from the grid, managing diversity through learned attention-based competition. This contributes a new path for the QD community, moving away from hand-engineered toward learned, continuous competition topologies.
>
> > I quote a claim of the authors: "A striking finding from our analysis is that even LQD (Q) variant trained purely for fitness optimization naturally maintain significant population diversity.". I think this conlusion is very tricky. [...] why LQD (Q) is worse than LQD (QD)?
>
> LQD (Q) is optimized *strictly* for maximum fitness. LQD (QD) is optimized for a mix of fitness and diversity. It is not "worse"; it is specialized for a different objective. The "striking finding" is that even when we tell the meta-learner "only care about the best fitness," it learns that maintaining diversity is an *instrumental goal* to achieve that fitness. This empirically validates the stepping stone hypothesis (Stanley & Lehman, 2015) via meta-learning.
>
> > (1) What is the dimension of the descriptor space (D) used in this paper? [...] why the relative distance could be maintained under the standard normal condition? prove it.
>
> For the BBOB tasks, we project the search space into a descriptor space of dimension $D=2$ for visualization and training simplicity. Regarding the preservation of relative distances: this is a direct application of the **Johnson-Lindenstrauss lemma**, which states that random projections preserve relative distances between points with high probability.
>
> > (2) what does the reproduction size B denotes?
>
> $B$ denotes the batch size of offspring generated at each generation (offspring added to the population before selection). We will clarify this definition in Algorithm 1.
>
> > (3) I can not agree with the authors the proposed complex neuroevolution training is efficient... use such huge computational resources to achieve competitive results with naive GA and MAP-Elite can not really prove the value of this work.
>
> It is crucial to distinguish between **meta-training cost** and **inference cost**. Training the meta-model is indeed expensive (8 GPUs), as is training any foundation model or meta-learned optimizer (e.g., AlphaZero, GPT). However, once trained, the **LQD algorithm itself is highly efficient**, running on standard hardware with negligible overhead compared to baselines. The value lies in the *discovered algorithm*, which can be deployed universally without re-training the transformer. Furthermore, we demonstrate that this discovered algorithm generalizes to robotic tasks it never saw during the expensive training phase.
>
> ---
>
> We hope these explanations clarify the distinct contribution of applying meta-learning to the specific problem of Quality-Diversity archives, and we are committed to updating the related work to reflect the broader MetaBBO field.

---

> ### Comment · Reviewer_Ziuf · 2025-11-21
> **further comments**
>
> I apprecaite the authors for their responses, which address some of my concerns, such as related work expansion and the other small questions. However, I have to remind the authors that in ICLR, during the rebuttal, you can revise your paper. However, I can not find any revision of your paper.
>
> On the other hand, I generally find the response perfunctory. Please read my review and answer my questions carefully.
>
> Most importantly, there are still two major concerns:
>
> 1) the authors claim that LQD(Q) could learn diversity concept naturally, the comparison is still limited, especially with latest MetaBBO works that use only quality indicator to train their approach. Besides, the comparison should also include training cost, exsiting MetaBBO typically require a small GPU (except works by Lange et al.) to achieve good results. Compare with latest ones such as GLHF[1] and RLDEAFL[2] to prove the technical advantage of your work. You have to compare exsiting baselines to tell readers your work is really good at generalization across problems and deserves that much training cost.
>
> [1] https://proceedings.neurips.cc/paper_files/paper/2024/file/19e9a88d91917775b34fdad447ed8908-Paper-Conference.pdf
>
> [2] https://dl.acm.org/doi/abs/10.1145/3712256.3726309
>
> 2)  Johnson-Lindenstrauss lemma indeed states that random projections preserve relative distances between points with high probability, what is the probability lowerbound in your setting?
>
> To summarize , I can not provide objective score update for now.

---

> ### Author Response · Authors · 2025-11-21
>
> We thank you for your continued engagement and for acknowledging our previous responses regarding related work.
>
> ---
>
> We have uploaded a revised PDF where we have carefully incorporated your feedback, particularly regarding the framing of related work and the technical clarifications discussed below.
>
> We address your two remaining concerns below.
>
> **1. Scope of Contribution and Comparison with MetaBBO**
>
> **Clarification on the Paper's Goal:** We believe there is a fundamental misunderstanding regarding the primary contribution of our work.
>
> * **We are not proposing a new MetaBBO method.** We explicitly state that we utilize an established MetaBBO framework (Lange et al., 2023) as our optimization backbone (see list at the end of our answer).
> * **We are proposing a novel Quality-Diversity (QD) framework (LQD).** Our contribution is the formulation of QD algorithms—specifically the archiving and local competition mechanisms—as a learnable problem.
>
> **Conclusion on Baselines:** Consequently, we respectfully maintain that a comparison to state-of-the-art MetaBBO methods, such as GLHF or RLDEAFL, is not relevant to the contributions of this paper. Our work does not aim to improve the meta-optimization process itself, nor do we claim novelty in the architecture of the optimizer. Instead, we apply existing MetaBBO techniques to a fundamentally different problem class: Quality-Diversity.
>
> **Clarification on Novelty:** This distinction regarding the scope of our work also directly addresses your initial concern regarding a perceived "lack of novelty." We believe this concern arose from viewing the work as a contribution to the MetaBBO literature rather than the Quality-Diversity literature. As stated, we do not claim novelty in the meta-optimization method itself. Instead, the novelty lies in the **LQD framework**: the specific parameterization that unifies disjoint Quality-Diversity concepts (like grid-based archives vs. unstructured archives) into a single, learnable attention-based architecture. This formulation allows us to search the space of QD algorithms for the first time, **which is a significant departure from existing hand-crafted heuristics, commonly used in QD algorithms.**
>
> ---
>
> **2. The Johnson-Lindenstrauss Lemma and Random Projections**
>
> You asked about the probability lower bound for the Johnson-Lindenstrauss (JL) lemma given our settings ($D=2$).
>
> **Response:**
> You are correct. The JL lemma provides bounds for embedding $N$ points into $k$ dimensions such that $k \sim O(\log N / \epsilon^2)$. For a target dimension of $k=2$, the distortion $\epsilon$ would indeed be large, and the strict theoretical guarantee of distance preservation does not hold strongly.
>
> **Action Taken:**
> Upon reflection, the claim that the projection preserves uniform distance is not necessary. We have **removed the claim** regarding preserving distance from the revised paper. The updated section now reads:
>
> > In standard QD settings, the evaluation process typically involves a mapping from genotype to phenotype, and subsequently from phenotype to descriptor. While this composite mapping is often highly complex and non-linear --- such as computing the trajectory of a robot from its controller parameters --- it nevertheless remains a function mapping the genotype to the descriptor. In this work, we simulate this mapping using a simple linear random projection from genotype to descriptor. Our empirical results demonstrate that meta-training on this simplified linear mapping is sufficient to learn competition rules that generalize to complex tasks, such as robot control tasks involving intricate trajectory-based descriptors.
>
> **Why this strengthens the paper:**
> In Quality-Diversity, the mapping from Genotype Space $\to$ Descriptor Space is, in general, **not isometric**. For example, in the Ant robot task, the descriptor is "feet contact" while the genotype is the neural network parameters. These spaces have completely different topologies and the mapping is **not isometric**; two completely different neural network controllers can map to the same feet contact descriptor.
> Therefore, using a random projection that *does not* perfectly preserve distances actually simulates the real-world difficulty of QD tasks better than an isometric projection would. It forces the LQD algorithm to learn robust competition rules that work even when the descriptor mapping is complex or non-linear. In this work, we show empirically that a simple linear mapping is enough to **generate a diverse set of tasks that allow the meta-learning process to discover generalizable local competition rules.**
>
> We are deeply grateful to you for raising this point, as removing this claim and clarifying the nature of the mapping has significantly improved the accuracy of the paper.
>
> ---
>
> We hope this clarifies the contribution of our paper and demonstrates that we have taken your technical feedback on the projection mechanism seriously.

---

> ### Comment · Reviewer_Ziuf · 2025-11-23
> **further comment**
>
> I raise my score from 2 to 4, a score of 6 requires the authors to further adjust their attitude, I do not think only adding the important related work in appendix is a respectful action, you should indicate them at main content. furthermore, I can not raise score to 6 due to the limited comparison. I can not agree with the author that proposing a concept is enough to get a ICLR paper, if so, this is unfair to other works that with solid experimental results. I hope the authors could understand my concerns.

---

> ### Author Response · Authors · 2025-11-25
>
> We thank **Reviewer Ziuf** for raising the score to 4. We want to address the remaining concerns to demonstrate that this work merits a clear acceptance.
>
> ---
>
> **1. Placement of Related Work (Action Taken)**: We sincerely apologize if placing the new references in the Appendix appeared disrespectful; this was certainly not our intention. The initial decision to place these references in the Appendix was driven strictly by the **rigid page constraints** of the main manuscript.
>
> - **Action:** To address this, we have revised Section 2 of the paper to explicitly highlight the importance of these recent works  and invite the reader to consult the Appendix for a comprehensive review of the state-of-the-art. We now clarify in the main text that we use the specific framework of Lange et al. for simplicity, acknowledging it as one tool among the broader, MetaBBO landscape.
>
> **2. Comparison against other MetaBBO methods**: We **strongly and respectfully disagree** with the premise that a comparison against state-of-the-art MetaBBO methods is required for this paper.
>
> As stated in our previous responses, we utilize MetaBBO strictly as a **tool** to discover our algorithm. **We do not propose a new MetaBBO method, nor do we claim that LQD is better than MetaBBO methods**; therefore, benchmarking our optimization backbone against SOTA MetaBBO methods is outside the scope of this contribution.
>
> - **Validation of the LQD Formulation:** The fact that we utilize the MetaBBO method from Lange et al. - rather than the latest SOTA meta-optimizer - actually **strengthens** our core claim. It demonstrates that the LQD formulation is so effective that it discovers superior QD algorithms even without state-of-the-art meta-optimizers.
> - **Comprehensive Relevant Baselines:** We have performed a comprehensive evaluation against the baselines that are relevant to our contribution: **MAP-Elites, DNS, GA, NS, and CMA-ES**. Our results demonstrate that the discovered LQD algorithms outperform these established methods. **We note that the other reviewers found this set of baselines appropriate and sufficient to validate our claims.**
>
> **Distinction between Optimizer and Architecture:** We view the distinction between the meta-optimizer (MetaBBO) and the discovered algorithm (LQD) as analogous to evaluating a new neural architecture (e.g., the Transformer) trained with a standard optimizer (e.g., SGD). **To prove the value of the Transformer architecture, one compares it against other architectures (e.g., LSTMs), not by benchmarking SGD against newer optimizers (e.g., Adam)**. Similarly, in our work, LQD is the new architecture (the contribution) and MetaBBO is simply the optimizer (the tool). Since the LQD algorithms (trained with standard ES) outperform established QD baselines (like MAP-Elites), the contribution is validated independent of the specific meta-optimizer used to find them.
>
> ---
>
> We wish to emphasize again that we **never** claim to beat existing MetaBBO methods or to introduce a new MetaBBO contribution. As a result, our manuscript **contains no unsupported claims regarding meta-optimization**. Our claims are strictly focused on the performance of the _discovered_ LQD algorithms against standard QD baselines, which are fully supported by our experimental results.
>
> We share the reviewer's interest in the rapidly evolving MetaBBO landscape. We are genuinely excited about the potential of combining the LQD framework with recent SOTA meta-optimizers. We believe that applying these advanced optimizers to the LQD search space is a promising direction that could yield even more powerful QD algorithms. However, for this foundational paper, our focus is on validating the LQD framework itself—demonstrating that archive mechanisms can be learned—rather than benchmarking the meta-learning process itself.
>
> We thank the reviewer for their time and hope this clarification warrants a re-evaluation of the score.

---

> > ### Comment · Reviewer_Ziuf · 2025-11-27
> > **final comments**
> >
> > I appreciate the authors for addressing my issue #1 to take respect on recent metabbo works and indicate them properly in the main content.
> >
> > I also understand the authors for why they did not compare LQD with MetaBBO approaches (maybe due to limited rebuttal time). However, understand can not mask the fact that in recent MetaBBO works, they often compare themselves with your backbone algorithm LES (at least they show respect to the LES by indicating it in related works and disccusion sections). Since all works including yours focus on solving optimization problems (black-box ones), I can not see why it is not a legal requirement to ask you add comparison. All I request is the respect and fair benchmarking to tell our future readers that what approach should be used under what cases. This is important, and also valuable results the readers expect from a good paper.
> >
> > I will give a 6 score for this paper considering that the authors have paid efforts in this rebuttal. However, I have to insist my position that, for this paper, I can live with it or without it, that means, I still think this paper is not good enough to arrive the acceptance side, yet I leave the final decisions to the ACs and PCs.

---

### Official Review · Reviewer_jRqg · 2025-10-22

**Soundness:** 4
**Presentation:** 4
**Contribution:** 2
**Rating:** 4
**Confidence:** 4

**Summary:**

This paper introduces a meta-learning approach that creates new
competition rules for QD algorithms. Essentially, the QD algorithm is framed as
based on genetic algorithms, where new solutions are generated and added to the
population, and the next generation is selected from that population via
competition. In this paper, the selection is handled by creating a transformer
that outputs ranking values for each solution in the new population; the top N
individuals are then chosen. During the meta-learning process, the transformer's
weights are trained so that the resulting QD algorithm optimizes a
meta-objective. Results on black-box and robot control benchmarks show that the
final meta-learned QD algorithms exceed the performance of existing QD baselines
in terms of both quality/fitness and diversity.

**Strengths:**

1. I find the idea of the paper quite novel among QD algorithms, even though it
   builds heavily on the single-objective evolutionary algorithms in Lange
   2023b. It is certainly rare to see a QD paper that involves meta-learning.
2. I appreciate the analysis of the learned local competition strategies in
   Section 5.2, as they help to understand how the variants operate after
   meta-learning.
3. The paper is overall well-written and easy to follow. It also has good
   statistical rigor, performing multiple trials and carrying out statistical
   tests to determine if differences in performance are significant.

**Weaknesses:**

The paper takes a rather narrow view of QD algorithms as only being based on
genetic algorithms. This view is stated on line 126 in the background, and it is
reflected in the choice of baselines --- MAP-Elites, NS, and DNS are all based
on genetic algorithms. However, QD in recent years has been able to reach a
large number of machine learning domains by growing far beyond its roots in pure
genetic algorithms. For example, works like PGA-MAP-Elites and PPGA incorporate
reinforcement learning (RL) mechanisms to scale to high-dimensional RL problems,
while BOP-Elites incorporates Bayesian optimization to improve sample
efficiency. Furthermore, for black-box problems, CMA-MAE has demonstrated
state-of-the-art performance on many difficult domains. In its current form,
this paper does not account for any of these advances. In order for this paper
to be relevant to a machine learning community like ICLR, I believe it needs to
demonstrate relevance to algorithms like the aforementioned. While it is true
that these works focus on different QD mechanisms (e.g., variation) than that
addressed in this paper, I believe they are important enough that this paper
should discuss how they integrate with the LQD framework.

For example, this paper could include CMA-MAE as a baseline or discuss how LQD
can be applied to CMA-MAE. Right now, I think a valid concern is that running
CMA-MAE simply exceeds any of the LQD variants presented here in both quality
and diversity. If that is the case, LQD does not seem very useful because one
does not need to run the expensive meta-learning process. Instead, they can just
run CMA-MAE immediately. However, this is no longer a concern if it is clear (and ideally, empirically proven)
that LQD can somehow integrate with and enhance CMA-MAE.

**Questions:**

1. Suggestion: The text in Figure 4 and Figure 6 is quite small and difficult to
   read. Also, LQD(D) is described first in the text despite being second in the
   figure, which I found confusing.
2. The robot control domains seem quite different from those of prior work. For
   instance, their genotypes are only several hundred parameters rather than the
   tens of thousands of parameters used in works like PGA-MAP-Elites. This makes
   it difficult to contextualize the results in this paper with those of prior work. It would be
   helpful to explain these differences somewhere.

---

> ### Author Response · Authors · 2025-11-20
>
> We thank **Reviewer jRqg** for the excellent summary and for recognizing the novelty of our meta-learning approach to Quality-Diversity. We appreciate the valid points regarding the broader QD landscape.
>
> ---
>
> > The paper takes a rather narrow view of QD algorithms as only being based on genetic algorithms... I believe they are important enough that this paper should discuss how they integrate with the LQD framework.
>
> We respectfully disagree with the characterization of our framework as taking a "narrow view." On the contrary, we argue that formalizing Quality-Diversity through the lens of Genetic Algorithms provides a **unifying theoretical framework** that is a strength, not a limitation. In scientific inquiry, the ability to unify diverse methods under a common formalism is a driver of progress.
>
> By viewing QD as a generalized Genetic Algorithm with local competition, we can decouple the **Variation/Emitter** mechanism (how offspring are generated) from the **Competition/Archive** mechanism (how survival is determined).
>
> 1.  **Unifying Modern QD:** State-of-the-art methods like PGA-MAP-Elites or PPGA fit perfectly within this unified framework. They essentially replace the standard Gaussian mutation with sophisticated gradient-based operator. They remain, fundamentally, evolutionary loops.
> 2.  **Orthogonality of Contribution:** The contribution of LQD is strictly on the **Competition/Archive** side. We aim to discover *how to select* survivors, regardless of *how they were generated*. In contrast, the works cited (PGA-MAP-Elites, CMA-MAE) innovate on the **Variation** side.
> 3.  **Choice of Baselines:** Because LQD focuses on the competition mechanism, we chose baselines that allow for a controlled comparison of *archives* (ME, DNS, NS) while keeping the *variation* (Gaussian mutation) constant. Comparing LQD (with Gaussian mutation) against CMA-MAE (with CMA-ES mutation) would confound the results, making it impossible to tell if gains came from the archive or the emitter.
>
> While LQD is theoretically compatible with gradient-based or covariance-based emitters (as they are orthogonal components in our unified framework), implementing these hybrid is outside the scope of this work, which focuses on establishing the LQD framework itself.
>
> > For example, this paper could include CMA-MAE as a baseline or discuss how LQD can be applied to CMA-MAE.
>
> While CMA-MAE is a strong baseline, it represents a different class of contribution (variation operator vs. selection operator). Comparing LQD (standard GA mutation) directly to CMA-MAE (covariance matrix adaptation) doesn't make sense here.
>
> > The robot control domains seem quite different from those of prior work. For instance, their genotypes are only several hundred parameters rather than the tens of thousands of parameters used in works like PGA-MAP-Elites.
>
> The controllers here are smaller, standard multi-layer perceptrons suitable for neuroevolution in this context. While PGA-MAP-Elites often uses larger networks because it leverages TD3/RL gradients, standard neuroevolution approaches (like the baselines we compare against) typically perform better with compact networks. We will clarify this context in the experimental setup.
>
> > Suggestion: The text in Figure 4 and Figure 6 is quite small and difficult to read. Also, LQD(D) is described first in the text despite being second in the figure, which I found confusing.
>
> We will increase the font size in all figures and align the text description order with the visual order in the final manuscript.
>
> ---
>
> We hope the clarification regarding the orthogonality of LQD (archive) and algorithms like CMA-MAE (emitters) addresses your main concern regarding the contribution's scope.

---

> > ### Comment · Reviewer_jRqg · 2025-11-21
> >
> > I thank the authors for their careful response to my comments. I agree with the authors that:
> >
> > - Variation and selection are separate components.
> > - CMA-MAE's main contribution is more on the variation side rather than the selection side.
> > - It is out of scope to try to develop a CMA-MAE variant that integrates with LQD.
> >
> > However, since CMA-MAE is the current SOTA, I maintain that it would be useful to include a comparison to CMA-MAE. Such a comparison would show where LQD stands with respect to it. If LQD does better, then that is great, and it shows the effectiveness of LQD. If it is worse, that is also fine -- it provides room to discuss how this difference occurs because CMA-MAE focuses on providing a powerful variation operator, while LQD provides a way to learn the selection operator. A combination of CMA-MAE and LQD could then be proposed as future work. Even if CMA-MAE is not run, I think this discussion should still be included -- I find it valid for a QD practitioner to ask "when should I use LQD vs. CMA-MAE?" Overall, my request aims at broadening the applicability of the paper.

---

> ### Author Response · Authors · 2025-11-21
>
> We thank **Reviewer jRqg** for their prompt and constructive response. We are pleased that we have reached a consensus on the fundamental distinction between the **Variation/Emitter** (CMA-MAE’s focus) and the **Selection/Archive** (LQD’s focus), and that combining them is outside the scope of this work.
>
> ---
>
> **1. On the request for an experimental comparison with CMA-MAE**
>
> We appreciate the reviewer’s suggestion to "show where LQD stands." However, we strongly believe that conducting this specific experiment - comparing a method with a sophisticated optimizer (CMA-ES in CMA-MAE) against a method with a simple Gaussian mutation (LQD) - would be scientifically misleading for the following reasons:
>
> * **Confounded Variables:** If CMA-MAE outperforms LQD (which is expected on BBOB functions, as CMA-ES is a superior optimizer to fixed-variance Gaussian mutation), it would not indicate that the LQD *archive mechanism* is inferior. It would merely confirm that CMA-ES is a better optimizer than Gaussian mutation. Conversely, if LQD were to outperform CMA-MAE, it would likely be due to noise or specific landscape properties, falsely suggesting that our learned archive compensates for a weaker optimizer.
> * **Narrative Consistency:** As the reviewer graciously noted, the outcome of this experiment **would not change the story of the paper**. A "loss" to CMA-MAE does not invalidate the finding that we can *meta-learn* QD algorithms, nor does it invalidate that LQD outperforms baselines using equivalent variation operators.
> * **Resource Allocation:** We respectfully urge the reviewer to consider that running new baselines on entirely different algorithmic foundations (requiring careful hyperparameter tuning to be fair) places a significant burden on authors during the short rebuttal window, **particularly when - as we agree - the results would not alter the paper’s core validity or claims**.
>
> **2. On the request for Discussion**
>
> We fully agree with the reviewer that a discussion on "when to use LQD vs. CMA-MAE" is valuable for the community. We are happy to include a dedicated paragraph in the final manuscript discussing this relationship.
>
> *Proposed addition:* While recent advances like CMA-MAE have significantly enhanced QD by integrating powerful variation operators (CMA-ES) to address search efficiency, our work focuses on the complementary challenge of the selection mechanism. We posit that meta-learning competition rules offers a distinct advantage when the primary bottleneck is not the optimizer, but the limitations of rigid archive structures.
>
> **3. The Bigger Picture:**  _Unifying QD and Escaping the Grid_
>
> We hope the reviewer will consider the broader impact of this work beyond benchmark leaderboards. The primary contribution of this paper is not to beat every existing hand-crafted heuristic, but to propose a paradigm shift: **moving away from rigid, hand-designed binning structures (grids) toward flexible, learned attention mechanisms.**
>
> For over a decade, QD has relied on the grid heuristic (MAP-Elites). LQD provides the first proof-of-concept that we can break this dependency using meta-learning, discovering algorithms that are open-ended, continuous, and adaptable. We believe this "unifying view" - treating QD as a generalized genetic algorithm with learned local competition - **opens a new research direction that is highly relevant to ICLR.**
>
> ---
>
> We hope this response addresses the remaining concerns. We stand by the rigor of our current experimental suite, which isolates the contribution of the *learned archive* by controlling for the variation operator.

---

> > ### Comment · Reviewer_jRqg · 2025-11-23
> >
> > I agree the discussion paragraph is sufficient and I am happy to give a positive score of 6.
> >
> > On a side note, I am a bit disappointed that the authors' comments appear to have been LLM-generated, e.g., the overall tone is too grandiose, there is unnecessary bolding of text, and there are awkward phrases like "Escaping the grid." I suppose it's not all bad, but I would have certainly preferred to hear from the authors themselves.

---

> ### Author Response · Authors · 2025-11-23
>
> We thank **Reviewer jRqg** for the continued engagement and for confirming that the proposed discussion regarding CMA-MAE and LQD is sufficient. We are glad we have reached a consensus on the scientific content.
>
> ---
>
> We wish to address the comment regarding the nature of our previous response directly. We were quite disheartened to read that our genuine enthusiasm for the project was interpreted as AI-generated text.
>
> We assure you that our previous response was not LLM-generated. The bolding was applied manually to improve readability, and the phrasing - including "escaping the grid" - reflects our genuine enthusiasm for moving the field away from fixed bins. Quality-Diversity algorithms promise to foster open-endedness, but we believe that the grid is the **opposite** of open-ended. As a result, we think that QD algorithms need to _break out_ of the grid.
>
> While we occasionally use LLMs for basic rephrasing or grammar checks, the arguments and tone are entirely our own. It is discouraging to hear our passion for the work interpreted as artificial.
>
> ---
>
> **Regarding the evaluation:** Since we have reached a consensus on the baselines and addressed the presentation issues, what specifically is needed to warrant a higher score?
>
> We believe this paper stands out because it does not simply propose a new variation operator (like many recent QD works). Instead, it targets the fundamental differentiator of QD algorithms: the selection/competition mechanism.
>
> Given the acknowledged novelty of this meta-learning approach and the rigorous evaluation across domains, we are genuinely curious: what specifically do you feel is still missing to merit a strong acceptance?

---

> > ### Comment · Reviewer_jRqg · 2025-11-23
> >
> > I thank the authors for their response. I should clarify that I am not too concerned about whether the response is LLM-generated or not, and my perception of that does not affect my score.
> >
> > Regarding the score of the paper, I agree that I have no further concerns. I have raised my score to a full accept (8).

---

> ### Author Response · Authors · 2025-11-25
>
> We thank Reviewer jRqg for the positive re-evaluation and for raising the score to an 8. We genuinely appreciate the constructive dialogue throughout this process.

---

### Official Review · Reviewer_DGYg · 2025-10-26

**Soundness:** 3
**Presentation:** 4
**Contribution:** 3
**Rating:** 4
**Confidence:** 3

**Summary:**

The authors propose a meta-black box optimization approach to discover new quality-diversity (QD) algorithms by encoding various competition rules with a Transformer architecture. They evaluate their approach in a function fitting optimization benchmark and in several simulated robot control benchmarks, showing that their approach matches or outperforms standard QD algorithms in diversity only, performance only or QD metrics.

**Strengths:**

- the motivation of the meta-QD approach based on biological evolution principles is well-written and quite convincing
- I could not find any typo

**Weaknesses:**

- some elements of the methods and some experimental results are unsufficiently clear (see below)
- a few claims are unsupported (see below)
- some figures have issues (see below)
- relevant variants and ablations are missing (see below)
- it would make sense to compare the approach with approaches which strive to learn a good descriptor space (e.g. AURORA and many others).

**Questions:**

** Questions: **
- would it be possible to analyse the learned transformer model to "decode" the competition rules learned by your LQD approach?
- The descriptor space dimension is D. How do you deal with different values of D in the architecture of Fig. 2?
- You generate a random matrix of size $R^{Dxn}$. What's n?
- p.5, you project high-dimensional genotypes into a descriptor space, but in most QD algorithms, the descriptor space is rather built on phenotypes, that is encodes a description of the trajectories performed by the agent. Did you try both types of encoding?
- Section 3.2.2 is unsufficiently clear. In table 1, what is the "Eval metric" column? It looks close to the raw score.
- Why does the Meta-objective use a median rather than a mean. Did you try variants? Same question about Tables 3 and 4.
- Why did you use SNES rather than any other ES? Did you try variants?
- Is the performance given in curves the average fitness of the final population? The fitness of the best individual? This should be clarified.
- In Section 4, you describe the hardware used, but not the optimization time. How expensive is it?
- In Section 4.2, you compare scores with respect to baselines, but you did not describe your efforts to make sure that the various algorithms benefit from the same computational budget. Is it the case? What is the training budget?
- Section 4.3 is missing a lot of experimental details. What are the 584 parameters for? It is "up to" 584, so which other sizes do you use in which contexts?
- do you see a possible alternative to using transformers in your architecture? Could you evaluate such a variant?
- I believe the topic of Appendix B.7 is important and should be moved to the main paper. BTW, the fact that "LQD leverages meaningful relationships between solutions captured be the descriptor space" is not supported by a detailed analysis. Again, providing such an analysis would strengthen a lot the paper. At least, can you provide an example of such a relationship?
- On figures 17 to 22, what are the axes?
- In Table 6, what is the descriptor space?
- In section E.3, the descriptor spaces should be described more accurately for readers who are not familiar with the QD literature

** Remarks: **
- In the baselines, you mention the Random baseline which does not appear in the results. You should specify immediately that this baseline is used only for performance normalization. I had to dig this information from the caption of Fig. 7, this is not the right place.
- Section 4.3 is "Robot control tasks", but there is no robot in your work, only simulations. You should turn any mention of "robot" into "simulated robot", as dealing with real robots is much harder (and probably you method cannot address this)
- Figures 4 and 5, normalization should be explained
- lines 435 sq.: the fact that LQD "creates crucial stepping stones" is not supported by any in depth analysis. Such analyses would strengthen the paper a lot
- Figure 6 does not have a title, nor a color scale, nor labels on the axes. Under its current form, it cannot be exploited for analyses. You need to correct this!
- Some text in Fig.7 is not readable.
- Appendices B1, B3, and E1 should be grouped into a single Section.
- You may make it more explicit that your meta-BB approach is trained on fitting functions, but evaluated on episodic control tasks, which makes it clearer that these evaluation tasks are completely OOD.
- I would like to see Figure 15 for more functions of the benchmark, as the ones you show could have been cherry-picked.


** Typos: **
- line 94, Section A -> Appendix A
- line 251: BBOB is not defined
- line 420: (Section 5.1), the -> and the

---

> ### Author Response · Authors · 2025-11-20
>
> We thank **Reviewer DGYg** for the detailed and insightful review. We are glad you found the motivation convincing and the presentation excellent. We have addressed your questions and remarks below.
>
> ---
>
> > it would make sense to compare the approach with approaches which strive to learn a good descriptor space (e.g. AURORA and many others).
>
> This is an excellent point. However, we view algorithms like AURORA as orthogonal to our contribution. AURORA focuses on *learning the descriptor space* (dimensionality reduction of observations), whereas LQD focuses on *learning the competition mechanism* within a given descriptor space. LQD is a new type of archive mechanism. In theory, one could combine AURORA (for descriptor learning) with LQD (for archive maintenance).
>
> > would it be possible to analyse the learned transformer model to "decode" the competition rules learned by your LQD approach?
>
> We attempted to interpret the learned rules in **Section 5.2 and Figure 6**, where we visualize the "competition landscape." This visualization shows that LQD (D) learns a distance-based repulsion (similar to novelty search), while LQD (Q) learns a fitness-sensitive attraction basin. Decoding the exact weights of the transformer is difficult due to the non-linear nature of attention mechanisms, but these visualizations provide the most interpretable proxy for the learned behavior.
>
> > The descriptor space dimension is D. How do you deal with different values of D in the architecture of Fig. 2?
>
> The raw inputs (fitness + descriptors) are projected into a fixed latent dimension $D_K$ (16 in our experiments) via the initial linear embedding layer. This allows the internal transformer layers to operate independently of the input dimension $D$.
>
> > You generate a random matrix of size . What's n?
>
> $n$ is defined in Section 2.1 and used throughout the background (Section 2) and the method (Section 3). $n$ represents the dimensionality of the genotype (solution) space.
>
> > p.5, you project high-dimensional genotypes into a descriptor space, but in most QD algorithms, the descriptor space is rather built on phenotypes, that is encodes a description of the trajectories performed by the agent. Did you try both types of encoding?
>
> We thank the reviewer for this insightful comment. In standard QD settings, the evaluation process typically follows the chain: Genotype $\rightarrow$ Phenotype $\rightarrow$ Descriptor. This entire process can be viewed as a complex composite function mapping the genotype to the phenotype.
>
> In our work, we emulate this composite mapping using a simple linear random projection. While this represents a simplified linear mapping compared to complex trajectory-based phenotypes, our empirical results demonstrate that it is sufficient for meta-learning robust competition rules. Notably, LQD trained on these simple projections generalizes successfully to complex, true phenotypic descriptors (e.g., robot trajectories) in our evaluation tasks, where it outperforms established baselines like MAP-Elites and DNS.
>
>
> > Section 3.2.2 is unsufficiently clear. In table 1, what is the "Eval metric" column? It looks close to the raw score. [...] Why does the Meta-objective use a median rather than a mean.
>
> We thank the reviewer for pointing out this ambiguity. The column **Eval metric** refers to the specific metric used to benchmark the performance of the discovered algorithms in the Experiments section (Section 4), which differs from the internal **Raw scores** used during meta-optimization (particularly for the QD variant). We have updated the table header and the accompanying text to explicitly clarify that this column represents the test-time evaluation metric.

---

> ### Author Response · Authors · 2025-11-20
>
> > Why does the Meta-objective use a median rather than a mean. Did you try variants? Same question about Tables 3 and 4.
>
> During preliminary experiments, we compared both mean and median aggregation for the meta-objective and observed no significant difference in the final performance. Consequently, we chose to rigorously follow the established methodology in [1], which utilizes the median. We maintained this consistency for the results reported in Tables 3 and 4 to ensure alignment with the training objective and standard benchmarks.
>
> > Why did you use SNES rather than any other ES? Did you try variants?
>
> We followed the precedent set by [1] for meta-learning evolutionary strategies. SNES is a standard, robust choice for high-dimensional neuroevolution [2]. We did not extensively tune the meta-optimizer as SNES performed reliably.
>
> In the past, we experimented with Sep-CMA-ES, with no difference in performance. SNES was marginally faster to run.
>
> > Is the performance given in curves the average fitness of the final population? The fitness of the best individual?
>
> The curves represent the **fitness of the best individual** found so far (for Quality) or the **average novelty** of the population (for Diversity). We will clarify the captions in the final version. The evaluation metrics are detailed in Table 1.
>
> > In Section 4, you describe the hardware used, but not the optimization time. How expensive is it? [...] I believe the topic of Appendix B.7 is important and should be moved to the main paper.
>
> The meta-training is computationally intensive (approx. 2 days on 8 H100 GPUs), but the *inference* (running the learned LQD) is very fast and comparable to standard baselines. We agree regarding Appendix B.7; if space permits, we will move the descriptor ablation to the main text as it confirms LQD relies on meaningful descriptor relationships.
>
> > In Section 4.2, you compare scores with respect to baselines, but you did not describe your efforts to make sure that the various algorithms benefit from the same computational budget. Is it the case? What is the training budget?
>
> We confirm that all algorithms (LQD and baselines) were evaluated using the exact same computational budget and population size to ensure a strictly fair comparison. Since LQD operates within the same standard evolutionary loop structure as the baselines (iterative generations of fixed population size), enforcing parity is straightforward.
>
> Specifically, all methods used a population size of $N=128$. For the BBO training tasks (Section 4.2, Figure 3), we used a fixed budget of **1 million evaluations**. We have updated Section 4.2 to make these experimental controls explicit.
>
> > Section 4.3 is missing a lot of experimental details. What are the 584 parameters for? It is "up to" 584, so which other sizes do you use in which contexts?
>
> The parameters refer to the neural network weights of the robot controllers (genotypes). The size depends on the specific robot morphology (observation/action space size). 584 is for the Ant robot; others like Hopper are smaller. All these small experimental details are given in Appendix.
>
> > do you see a possible alternative to using transformers in your architecture? Could you evaluate such a variant?
>
> We thank the reviewer for this interesting suggestion. The choice of the Transformer architecture was driven by the requirement for **permutation equivariance**—ensuring that the competition outcome for any given individual depends on the population context but remains independent of the specific ordering of individuals.
>
> We acknowledge that other architectures, such as the Perceiver, also satisfy this property. In fact, we conducted preliminary experiments with Perceiver-based variants to potentially reduce computational complexity. However, we found that they were significantly more difficult to optimize via Meta-BBO compared to the standard Transformer.
>
> Since the primary contribution of this work is the **LQD framework** itself—demonstrating that competition rules can be meta-learned—rather than the specific neural parameterization, we prioritized the robustness of the Transformer. We believe that a comprehensive ablation study of different set-processing architectures is orthogonal to our main claims and falls outside the scope of this paper.

---

> ### Author Response · Authors · 2025-11-20
>
> > On figures 17 to 22, what are the axes?
>
> We thank the reviewer for pointing out this omission. The axes in Figures 17 to 22 represent the two dimensions of the descriptor space used for each specific robot control task (e.g., feet contact patterns or velocity components). We have updated the captions for these figures to explicitly state what the axes represent.
>
> > In Table 6, what is the descriptor space?
> > In section E.3, the descriptor spaces should be described more accurately for readers who are not familiar with the QD literature
>
> Regarding the BBO tasks (Table 6 context), we have updated Appendix A.1 to explicitly state that, unlike robotic tasks with physical behaviors, these mathematical benchmarks use a synthetic descriptor space generated via random projection of the genotype, as established in our Method section.
>
> Regarding the Robot Control tasks (Section E.3), we have expanded the description to provide precise definitions of the behavioral descriptors used. We now explicitly define "Feet Contact" as the proportion of ground contact time for each limb (capturing gait) and "Velocity" as the average velocity vector, ensuring the text is accessible to readers outside the specific QD niche.
>
>
> > In the baselines, you mention the Random baseline which does not appear in the results. [...] Figures 4 and 5, normalization should be explained.
>
> We apologize for the confusion. The Random baseline is used solely to normalize the scores in the bar charts (Figures 7 and 8) to make them comparable across different tasks. In Figures 4 and 5, normalization is done relative to the MAP-Elites baseline (where MAP-Elites = 1.0). We will make this explicit in the captions.
>
> > lines 435: the fact that LQD "creates crucial stepping stones" is not supported by any in depth analysis.
>
> Our claim regarding stepping stones is inferred from the results in Figure 5: LQD (Q) maintains significantly higher diversity than a standard GA while achieving higher fitness. This implies it preserves diverse, initially sub-optimal solutions that later lead to higher peaks, which is the definition of the stepping stone principle.
>
> > You may make it more explicit that your meta-BBO approach is trained on fitting functions, but evaluated on episodic control tasks...
>
> We agree this is a key strength. We will emphasize that the robot control tasks are completely out-of-distribution (OOD).
>
> We will implement your suggestions regarding Figure labeling, grouping appendices, and correcting typos. We hope these responses clarify the method and demonstrate the robustness of our work.
>
> ---
> We firmly believe that there is no unsupported claim in this paper, so we hope that our rebuttal has clarified these misunderstandings.
>
>
> [1] Discovering Evolution Strategies via Meta-Black-Box Optimization, https://arxiv.org/abs/2211.11260
>
> [2] NeuroEvoBench: Benchmarking Evolutionary Optimizers for Deep Learning Applications, https://arxiv.org/abs/2311.02394

---

### Official Review · Reviewer_G9q5 · 2025-10-31

**Soundness:** 3
**Presentation:** 4
**Contribution:** 3
**Rating:** 4
**Confidence:** 3

**Summary:**

Quality diversity (QD) algorithms are a relatively new class of evolutionary algorithms which focuses on maintaining population diversity by constraining competition between individuals to similar individuals. Several QD algorithms have been proposed in the past, including MAP elites (ME) and dominated novelty search (DNS).

The paper proposes to meta-learn QD algorithms (a transformer architecture) by learning competition rules between individuals from black-box optimisation tasks. The method consists of an inner, evolutionary loop where a given fixed QD-model is applied for some generations, and an outer-loop that optimises models based on the outcome of the inner loop.

The learned QD-algorithms are compared empirically with classical QD-approaches, such as ME and DNS, showing favourable performance, and generalisation ability.

**Strengths:**

Meta-learning of evolutionary algorithms have been around for some time. The approach has been evaluated on a large number of problems, ranging from the classical BBOB benchmarks to robot controller tasks. The learned QD approach partly outperforms ME and DNS on the BBOB benchmarks.

**Weaknesses:**

The application of statistical methodology could have been better. For example, in table 4, it would be useful to see p-values from statistical tests, or some information about variance.

There is no proper analysis of scalability. I would have liked to see how the performance scales with increasing problem dimension across all methods considered for some problems.

There is a very limited ablation study. It would be particularly interesting to understand the impact of the transformer architecture on performance. The network is tiny. How much performance improvement would a large network provide?

The paper does not compare with state of the art black-box optimisation algorithms. To give a clear idea about performance, I would expect to see a comparison with CMA-ES.

**Questions:**

How does LQD compare with CMA-ES?

What is the scalability of the approach?

---

> ### Author Response · Authors · 2025-11-20
>
> We thank **Reviewer G9q5** for their positive assessment of our work, acknowledging the favorable performance and generalization ability of our approach. We appreciate the constructive feedback regarding statistical analysis and comparisons.
>
> ---
>
> > The application of statistical methodology could have been better. For example, in table 4, it would be useful to see p-values from statistical tests, or some information about variance.
>
> We would like to highlight that our experiments are rigorously replicated with **32 independent random seeds**. As stated in Section 4, we assessed statistical significance using the Wilcoxon-Mann-Whitney U test with Holm-Bonferroni correction.
>
> In the current manuscript, we report p-values in the text (e.g., LQD (Q) vs GA $p < 10^{-10}$).
>
> To follow your advice, we have updated Table 2 to 4 to explicitly include median and **interquartile range**.
>
> > There is no proper analysis of scalability. I would have liked to see how the performance scales with increasing problem dimension across all methods considered for some problems.
>
> We thank the reviewer for this comment, which highlights that our scalability analysis was not sufficiently prominent in the main text. **We actually provide an analysis of scalability in Appendix B.6 (Figure 15)**.
>
> This figure demonstrates the generalization of our learned algorithm to larger population sizes (up to 1024) and problem dimensions (up to 32), despite being trained on populations of 128 and dimensions between 2 and 12. The results show that LQD maintains or increases its advantage over MAP-Elites as dimensions increase. We will make sure to reference this analysis more prominently in the main text.
>
> **We have now added an analysis of scalability for all variants.**
>
> The main robot control experiments in the paper are also using a larger number of dimensions (up to 584) than the BBOB tasks LQD was trained on.
>
> > The paper does not compare with state of the art black-box optimisation algorithms. To give a clear idea about performance, I would expect to see a comparison with CMA-ES.
>
> We understand the inclination to compare with CMA-ES, as it is indeed a state-of-the-art Black-Box Optimizer (BBO). However, it is important to note that CMA-ES and LQD belong to fundamentally different classes of algorithms. CMA-ES is an Evolution Strategy (distribution-based) focused exclusively on single-objective optimization (fitness), whereas LQD is a Quality-Diversity algorithm (population-based) designed to maintain a diverse archive of high-performing solutions.
>
> Comparing them directly is not "apples-to-apples": CMA-ES is expected to converge faster to a single peak but will fail to capture the diversity of the search space, which is the primary goal of QD algorithms. However, to provide the requested context, **we will update the Pareto front in Figure 3 to include the CMA-ES data point**, illustrating its high fitness but near-zero diversity compared to LQD and other QD baselines.
>
> > There is a very limited ablation study. It would be particularly interesting to understand the impact of the transformer architecture on performance. The network is tiny. How much performance improvement would a large network provide?
>
> The compact network size (6 layers, 16 features) was a necessary design choice because, unlike gradient-based training, Meta-BBO suffers significantly from the curse of dimensionality as parameter counts increase. Despite this, our results show that even this lightweight architecture outperforms human heuristics. Given the high computational cost and limited space, we prioritized other critical ablations, such as analyzing the impact of descriptors (Appendix B.7), over scaling the architecture.
>
> **Questions:**
> > How does LQD compare with CMA-ES?
>
> See updated Figure 3 in the paper.
>
> > What is the scalability of the approach?
>
> As detailed in Appendix B.6, LQD scales remarkably well. Despite being meta-trained on low-dimensional problems ($D \le 12$), it generalizes effectively to robot control tasks with hundreds of parameters and to larger population sizes without retraining.
>
> ---
>
> We hope these clarifications address your concerns, particularly regarding the statistical rigor and the distinction between pure BBO and QD. Would these explanations and the proposed updates to the tables/plots be sufficient to raise your score?

---

> ### Author Response · Authors · 2025-11-27
>
> We would like to politely verify if you have had the opportunity to review our responses and the updated manuscript.
>
> ---
>
> We believe we have effectively resolved your three concerns regarding **statistical rigor**, **baselines** and **scalability**:
> * **Statistical rigor**: In table 2 to 4, we are now showing median with **interquartile range**.
> * **Comparison with CMA-ES:** We have updated **Figure 3** to include CMA-ES. This explicitly illustrates that while CMA-ES achieves high fitness, it offers near-zero diversity compared to LQD, highlighting the distinct value of our approach.
> * **Scalability Analysis:** We have added more experiments in **Appendix B.6**. We evaluated all LQD variants (Q, D, and QD) on 6 BBO functions with dimensions varying in [2, 4, 8, 16, 32] and population size varying in [64, 128, 256, 1024].
>     * These results demonstrate robust generalization well beyond the meta-training parameters (dimensions 2–12, population 128).
>     * This complements the robot control results in the main paper, which further demonstrate scalability to high-dimensional and out-of-distribution problems.
>
> ---
>
> We hope that these additional results fully address your questions and that the substantial improvements to the paper warrant a re-evaluation of your score. Thank you again for your time and constructive feedback.

---

### Author Response · Authors · 2025-11-20

We thank the reviewers for their insightful comments and constructive feedback. We are encouraged that reviewers found our motivation "well-written and quite convincing" (**Reviewer DGYg**), our approach "quite novel" (**Reviewer jRqg**), and our presentation "excellent" (**Reviewer G9q5**, **Reviewer jRqg**). Before addressing specific questions, we wish to clarify three common points raised regarding baselines, scope, and statistical rigor.

1. Comparison with CMA-ES: **Reviewer G9q5** and **Reviewer jRqg** suggested comparisons with CMA-ES or CMA-MAE. We wish to clarify the scope of our contribution.
    - **CMA-ES vs. LQD**: CMA-ES is a specialized Evolution Strategy (distribution-based) designed for single-objective continuous optimization. In contrast, LQD is a Genetic Algorithm (population-based) designed for Quality-Diversity (QD). While we expect CMA-ES to reach higher peak fitness values due to its specialized nature, it is not a QD algorithm and does not maintain a diverse archive. Comparing them directly is an "apples-to-oranges" comparison of optimization paradigms (ES vs. GA). Nonetheless, we have updated the pareto plot (Figure 3) with results for CMA-ES. The novelty of CMA-ES is actually 0, so we draw a vertical line to show the fitness value.
    - **CMA-MAE vs LQD**: We agree that CMA-MAE is a state-of-the-art QD algorithm. However, there is a fundamental architectural difference. In standard QD (like MAP-Elites, PGA-MAP-Elites, or DCRL), the emitter (variation) and the repertoire (archive) are relatively independent components. In contrast, CMA-MAE tightly couples the emitter and the repertoire, as the emitter's adaptation and behavior are dictated by dynamic repertoire thresholds. LQD focuses on discovering a new archive/selection mechanism for the "independent" class of QD methods. While exploring how LQD containers can be combined with the tightly-coupled logic of CMA-MAE is an interesting line of research, we believe this is out of scope for this first paper, which aims to demonstrate that new QD containers can be discovered by meta-learning for traditional QD methods.
2. The "Narrow View" of QD and Orthogonality: **Reviewer jRqg** noted that the paper focuses on GA-based QD and omits RL-based or Gradient-based QD methods (e.g., PGA-MAP-Elites, DCRL-ME).
    - **Clarification**: This distinction arises because LQD introduces a novel archive and selection mechanism (how solutions compete). This is completely orthogonal to the variation operator (how solutions are generated). Algorithms like PGA-MAP-Elites or DCRL-ME improve performance by using sophisticated variation operators (gradients/RL) but still rely on the standard grid-based archive for competition.
3. Statistical Rigor **Reviewer G9q5** and **Reviewer Ziuf** commented on statistical validation.
    - **High Rigor**: We emphasize that all our experiments are replicated with 32 independent random seeds. This is significantly higher than the standard in the QD and RL literature (which often uses 5 to 10 seeds), providing a robust estimate of performance.
    - **Significance**: As noted in Section 4, we assess statistical significance using the Wilcoxon-Mann-Whitney U test with Holm-Bonferroni correction. We will update Table 4 and relevant sections to explicitly include p-values and variance information to make this rigor more visible.

---

### Meta-Review · Area_Chair_cHzL · 2026-01-09

**Summary:**

G9q5 points out limited scalability and no ablation study; DGYg points the lack of clarity of experiments, some unsupported claims, the lack of ablation studies; jRqg had specific GA concerns; Ziuf points out the lack of novelty and problems in validation.

**Reviewer Concerns:**

The rebuttal and discussions led to an increase of some scores, but at the same time, the authors' attitude in discussions was pointed out (Ziuf). Indeed, the authors do not hesitate to ask questions whose intended motive is to reach a specific score ("we are genuinely curious: what specifically do you feel is still missing to merit a strong acceptance?", to jRqg; "We thank Reviewer Ziuf for raising the score to 4. We want to address the remaining concerns to demonstrate that this work merits a clear acceptance." to Ziuf).

**Reviewer Scores:**

It is unclear, given the authors' somehow aggressive strategy to chase substantial increase in score (see above).

---

### Decision · Program_Chairs · 2026-01-26

Reject